# Can moral reasoning be modeled in an experiment?

Ján Grác[1], Adam Biela[2], Piotr Janusz Mamcarz[3]*, Dorota Kornas-Biela[4]

**1** Department of Psychology, Trnava University, Trnava, Slovakia, **2** Department of Social Psychology, Maria Sklodowska-Curie University, Lublin, Poland, **3** Department of Emotion and Motivation Psychology, The John Paul II Catholic University of Lublin, Lublin, Poland, **4** Department of Psychopedagogy, The John Paul II Catholic University of Lublin, Lublin, Poland

* pmamcarz@kul.pl

**Data Availability Statement:** The datasets generated during and/or analysed during the current study are not publicly available due to containing sensitive and qualitative information, so are available from the member of Trnava University

## Abstract

A review of the literature on moral issues indicates that none of the empirical approaches to moral reasoning proposes an experimental approach which controls for such object-related experimental variables as: knowledge, motivation, acceptance of moral norms and consequences of human behavior in moral situations in a single research procedure. A unique element of the proposed experimental method is a multi-stage model determining morality indicators. In the two-phase design experiment, psychology students were asked to create model ethical stories and then conduct an overall assessment of each of these stories. As a result, a base of ethical stories was created with empirical moral indicators (positive, negative, neutral). The patterns in the moral evaluation of ethical stories were determined by identifying three processes (selection, differentiation and integration). The final result is a confirmed design of the experiment and a set of formulas that can be used in education and research on morality reasoning.

## Introduction

In our study, we present an experimental approach to the study of moral reasoning which allows for the control of some indicators of morality as independent variables. A research proposal for this new approach was first presented by the Slovak educational psychologist Jan Grác. This proposal will be presented first in the context of the classic approaches to moral reasoning, and then in the latest contemporary research, with particular emphasis on the opportunities for experimental exploration of this research field [1].

Empirical studies on moral reasoning are well known in the psychological and education al literature [2–6]. They are inspired by the two early concepts of moral reasoning offered by Piaget [7, 8] and Kohlberg [9, 10] who do not use experimental methods in their research. In addition, in a meta-analysis of empirical studies on moral reasoning studies published from 1940 through 2020, the authors [11] indicate that studies on morality have largely neglected to examine how moral reasoning relates to actions in experimental contexts, including moral norms and intentions.

Ethical Board – professor Marián Špajdel (marian.
spajdel@truni.sk) on reasonable request.

**Funding:** This research received no specific grant
from any funding agency, commercial or not-for-
profit sectors.

**Competing interests:** None

## Classic approaches to moral reasoning

Piaget [7, 8] began his research on moral development using the clinical-experimental
approach. Piaget's research approach was a kind of methodological revolution, which shifted
psychologists' orientation from analyzing an individual's behavior or choice to the underlying
form and structure of moral reasoning. This approach was based on short dialogues with indi-
vidual small children in the context of a moral issue situation. The dialogue with a child, called
an *indagation*, aimed to ascertain his or her individual understanding of moral concepts. Pia-
get proposed a model of stages of moral reasoning which states as one of its premises that qual-
itative changes in the structure cause the appearance of a new stage at specific points in time.
Piaget created a non-quantitative but qualitative and creative method of scientific exploration
of moral thinking in young children.

The creative value of Piaget's clinical experiment was used by Grác [1, 12] to create a second
phase of the experiment called *modeling* and *concluding ethical stories*, conducted by students.
The methodological achievements of Piaget and Grác can be seen as analogous to the path that
led Dewey [13] in his time to discover the five-phase structure of human solution-oriented
thinking, which launched the empirical psychology of thinking with experimental research.

Another approach developed by Kohlberg [9, 10] is based on hypothetically constructed
moral dilemmas in order to test moral decision-making in youth and adults. A significant con-
tribution with regard to Kohlberg's studies was made by Rest [2, 14], who invented a proce-
dure called the Defining Issues Test (DIT). The test requires a subject to read a hypothetical
moral dilemma and then select those statements that are the most important issues in making
a decision about the test-case statements. According to Gibbs [15], Czyżowska and Niemc-
zyński [3], Kohlberg traced the development of moral structures through a series of six stages
which form an invariant and culturally universal sequence. Moral structures are measured by
requesting individuals to reason about cases in which two or more conventional values are in
conflict. The subject reacts to a moral dilemma, indicating what ought to be done and justify-
ing this course of action. The interviewer tries to elicit and probe the subject's views without
interjecting or suggesting any thinking different from the subject's own spontaneous thinking.
Recently, new authors in psychological investigations of moral reasoning have frequently
employed moral dilemmas [16, 17] where the subjects' behavior is interpreted in terms of com-
petence levels of moral reasoning.

However, analyzing the logical foundations of Kohlberg's research paradigm (what can be
seen, e.g. [18–20]) on hypothetical moral dilemmas, one can draw the conclusion that research
on moral evaluation is itself dilemmatic, primarily because of the one-dimensionality in which
the dilemmas in moral judgement literature continue to be formulated. In Kohlberg's
approach psychologist [19], could not apply the experimental procedure in a strict manner so
that the moral norms present in the tests of dilemmas would be controlled as independent vari-
ables having a hypothetical influence on the moral reasoning contained in the psychometric
tests. In this type of research on moral reasoning, a researcher choses to expose the subjects to
standardized tests of moral dilemmas, where the stimuli and test conditions had to remain
constant by definition (*ex defintione*). In the experimental study based on Grác's methodology
[1, 12], ethical stories will be presented not as a one-dimensional test reality but as a multidi-
mensional moral context, which allows us to take a new research approach to moral reasoning.
Thus, in our study the indicators of morality as an important context of moral evaluation will
be treated as independent variables, the influence of which are strictly controlled by the
experimenter.

Our most recent meta-analysis of the literature on moral reasoning from 1919–2020 found
only three articles with some connection to the experimental framing of the research question.

However, only one of these papers was based on laboratory-type experimental research [21], while the other analyses were based on: a thought experiment [22] and an online experimental procedure [23]. In all the above-mentioned authors the research methodology includes the paradigm of moral dilemmas. In this context, let us pose the question: To what extent is it possible to use the methodology of laboratory experiment to study moral reasoning having as an example of experimental research by Sudić & Ćirić?

Sudić & Ćirić [21] designed the laboratory experiment of 3x2x2 within groups, where the independent variables were: a) content of norm (moral, conventional, or abstract), b) rule type (obligation, or permission), and c) induced dilemma (punishment dilemma, or reward dilemma). Three dependent variables were measured in the experimental task: response time, accuracy and final confidence of performance. One of these main experimental findings is that moral rules are easier to process than conventional ones, and conventional rules are easier than abstract ones. However, these authors indicate that they are unsure as to what degree conventional and abstract rules contaminated the moral ones. Moreover, this approach is limited because the 'correct' response defined by this experiment is only logical consistency, and it does not take into account the broader context of real life moral reasoning.

Such experiment can be treated as a good initial exploration of the problematic of multidimensional field of moral reasoning, which is difficult to study using the method of a laboratory experiment where a possibility of control variables is rather too limited.

The second experimental approach to moral reasoning presented in our meta-analysis is a very intriguing thought experiment by Rhim, Lee & Lee [22], who studied human moral reasoning types in an autonomous vehicle moral dilemma, in a cross-cultural comparison of Korean and Canadian subjects. These authors could enrich our knowledge about the possibilities of simulation studies in the field of artificial moral intelligence regarding descriptive models of drivers' moral reasoning in problematic road traffic situations (e.g. Tunnel Problem [24]; MIT Moral Machines [25]). However, the study of moral reasoning in road traffic situations is not exhausted by dilemmas, however. It would be useful to consider the behavior of real drivers to a greater extent, to take into account moral indicators as factors influencing complex moral thinking processes. However, this type of research, while it is referred to as *thought experiments*, is not geared towards controlling independent variables as in typical experiments. The methodological goal here is to develop the most accurate descriptive model of some unitary situation of human behavior under specific conditions. Thought experiments are simply case studies of analyzed situations, where the control of variables is very limited. Hence their limited usefulness for experimental studies of moral reasoning, which is a matter of formulating general conclusions. It is these more complex processes subject to manipulation in Grác's experiments which are featured in our study.

Authors of a third study [23] presented an online experiment investigating the influence of the market environment on moral reasoning. A control group of respondents was asked to make a choice regarding moral dilemmas, while the study participants were exposed to either a market or non-market environment, and after that were asked to solve moral dilemma experimental tasks. While in principle online experiments are not objectionable, our main objection to this research is the fact that it does not consider morality indices as independent variables at all.

In the context of the above statements, one can pose the question: Is it possible to design an experiment on moral reasoning in circumstances of higher ecological validity, and if so, in what way?

## J. Grác's theoretical and methodological statements on moral reasoning

Unfortunately, none of the above approaches to moral reasoning propose any form of research which controls for object-related experimental variables, such as those related to motivation, acceptance of moral norms, and consequences of human behavior in moral situations–in a single research procedure. Grác states that this is possible, and we will demonstrate this in our experiment conditions, where subjects had as a task to define their real-life moral issue as a situation characterized by moral norms involved with the social-moral environment of their professional or educational endeavor. Such arrangements guarantee appropriate circumstances for significantly higher ecological validity of the experimental procedure.

Moreover, Grác [1] assumes that moral reasoning is a process where the subject conducts a moral evaluation of a human event based upon some premises. The premises for him or her are the indicators of morality which are analyzed in the particular situations, and then synthetized as an overall moral assessment of their activity (or non-activity) under the given social environment. Thus, a quite intriguing research question is: What kind of strategies do people use in the process of integrating indicators of morality in a particular moral situation into an overall moral evaluation of the human activity that is assessed?

This novel approach to the evaluation of moral reasoning processes is proposed by J. Grác, who has made a conceptual contribution towards the psychological analysis of the interiorization processes of moral norms. It is worthy of note that Grác in his original Slovak work [1] distinguishes between the terms: *mravnost'*, and *moralnost'*. In the *Introduction to the Polish edition* of Grác's book, Biela [12] emphasizes that these two terms only exist in discriminative usage in two contemporary languages, i.e., Czech and Slovak, and the term *mravnost'* is no longer used in Slavic languages other than Czech and Slovak. What differentiates the meanings of these two terms?

Grác on page 71 states that the term "mravnost'" denotes a morally good performance as the highest ethical level of evaluation [1]. It can be said that this is so because this term indicates the connection of morally good behavior with the accompanying positive psychological experiences, whereby the moral norm regulating both compounds of the conjunctive relationship: morally good behavior and the accompanying morally good experience–indicate a deep interiorization of the moral norm constituting the psychological basis of this conjunction. In turn, the term "moralnost'" according to Grác means the externally observed manifestations of morally good behavior without accompanying psychological experiences. Hence, not all performances observed as morally good are simultaneously "mravne" (i.e., moral), but only those where good performance is associated with morally good psychological experiences. Therefore, "mravnost'" is interpreted by Grác as an immanent psychological phenomenon.

What is the psychological novelty of Grác's conception of morality, and what is its methodological significance for an experimental approach to moral reasoning processes? Its greatest novelty is in extending the conditions of assessment of human performance in the defined situation as morally good or bad. In order to define the criteria enabling such an evaluation, Grác distinguished two parallel analytical dimensions of human performance: (1) moral behavior, i.e. the external revealing of moral decision making (external dimension); (2) the internal experience related to the moral behavior, i.e. intentional focus on the moral norm, motivation (internal dimension). In this way, the psychological understanding of moral norms as factors regulating human performance is enhanced by experiential and behavioral structures, creating a basis for systematizing norms as moral, executive, and legal. Executive and legal norms regulate only externally revealed human performance. The characteristic of moral norms is that they regulate both human behavior associated with internal experiences focused

on a particular moral norm as well as these internal experiences themselves (i.e. internal experience not tied with revealed behavior [1].

In this context, we can evaluate an individuals' ethical conduct in a specific situation as being a morally good action or a morally bad one. However, *morally good* vs. *morally bad* designates human behavior in a concrete situation, independent of whether the specific moral norm, e.g., hard work, courage, honesty, respect, trustworthiness, dedication, loyalty, has been actualized or not. Grác analysis posits that a person who accepts a particular moral norm but does not meet it could not be evaluated as performing a morally good action. And, similarly, a person could not be evaluated as having performed a morally good action if that person behaved in a manner consistent with the specific moral norm yet he or she did not accept this norm. Only the conjunction of the positive internal experience of the moral norm with the positive fulfillment of this norm in the situation provides the necessary and sufficient conditions for the positive ethical evaluation of a person acting in a given situation.

The mental processes involved in ethical evaluation of moral performance in everyday life situations and, particularly, in extreme situations are not passive ones on the part of the human subject; they involve active interaction with the social and natural environment.

Grác [1, 12] states that moral reasoning implies two sets of mental processes related to moral norms. The first set is related to the mental interiorization of moral norms, and the second set deals with the regulative function of moral norms in an individual's personal life, social interactions, and the natural environment. These mental processes create real-life moral situations, where moral norms interact with executive and legislative norms. Grác [1, 12] presents the internormative extensional relations between executive, moral, and legislative norms using Venn diagrams (Venn, 1884), what can be seen in Fig 1. As a justification of the purposefulness of using Venn diagrams in our analysis we can state that there is the need to prevent inaccuracies resulting from the extensional crossover of the norms under consideration. Such inaccuracies are indicated by Sudić & Ćirić [21] who, interpreting the results of their experimental research on moral reasoning, pointed out that they are unsure to what degree conventional and abstract rules contaminated moral rules in their experiment. Such contaminations, invalid and unreliable interpretations of the obtained results of empirical studies, can be prevented by precisely established scope relations between the designations of names of concepts, rules and norms presented in Venn diagrams.

## Materials and methods

This research was approved by The Department of Psychology Ethical Board (Trnava University). The data were analyzed anonymously but the oral consent was obtained before the subject research.

The Venn diagram illustrated in Fig 1 demonstrates that if the logical point of view is taken into consideration, we can see the relations between the three universal classes of norms: executive (E), moral (M), and legislative (L); thus, we can dislocate the extensional connections into the seven separate subclasses of norms. Both the executive and legal norms are described in Grác's [12] classification as beyond moral ones.

What characterizes executive norms? According to Grác [1] they are unambiguous regulators of the behavior manifested in specific situations, e.g., in contact with nature, with other people, with oneself. They are oriented towards the *hic et nunc* point of view, which is the optimum performance of certain activities. For example, in a sport, a sports record can be quickly replaced by another sports record (ski jumping hill record is marked in green to be visible to the jumpers). Another clear feature of this type of normative regulations is that although they improve human activities and accompany various mental experiences (e.g., attitudes, motives),

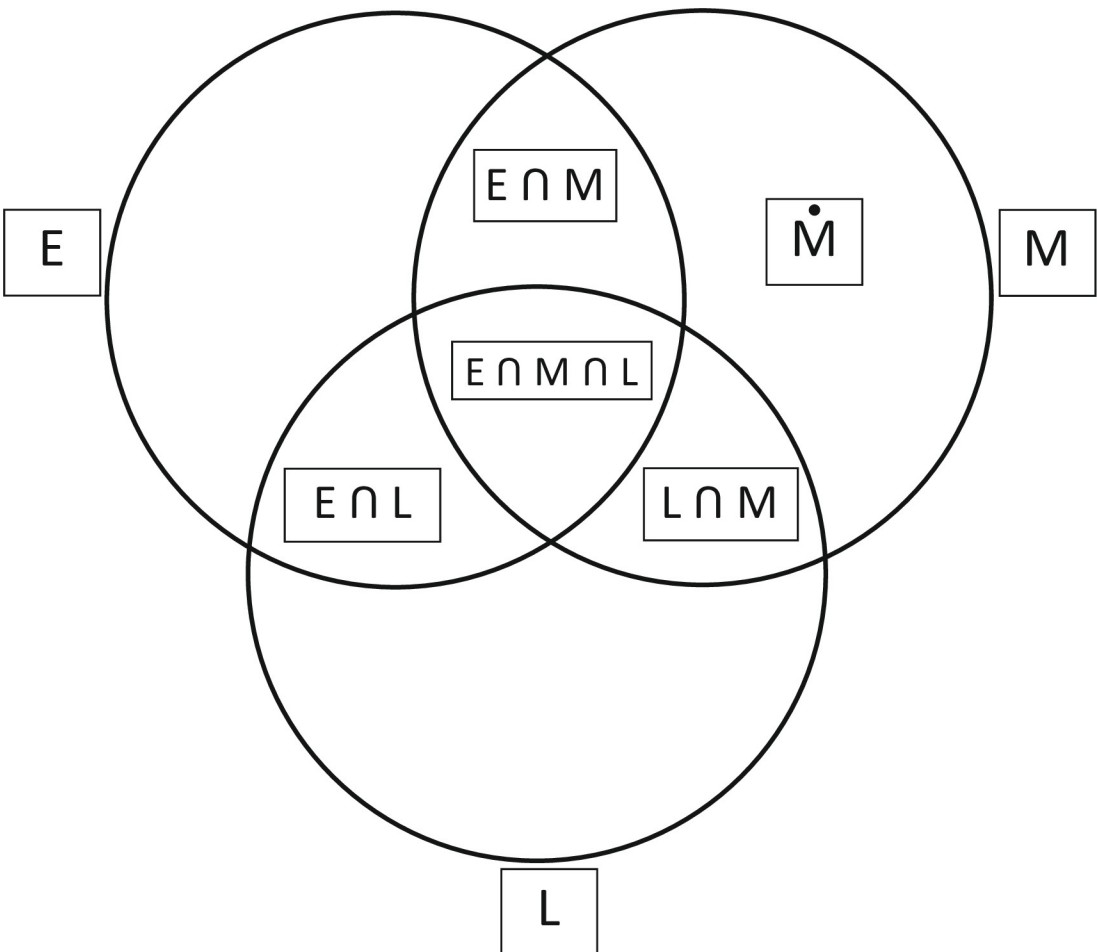

**Fig 1. The internormative extensional relations between executive, moral and legislative norms.**

the content of these experiences themselves is not subject to assessment by these norms. Thus, it can be said that neither the views nor the feelings accompanying the performance of these activities are subject to regulation within the framework of moral norms.

What do legal norms refer to? Grác [1] indicates that the subject of their regulation is also the psychological dimension of the behavior manifested, which is characterized by strictly defined effects. The specificity of these terms is that from the point of view of an injunction and prohibition, they strictly and unequivocally demarcate not only the lower limit of the effect (which is still allowed), but also the upper limit of this manifestation (which is already prohibited). The dimension of mental experiences accompanying the behavior, on the basis of a subjective cause, makes it possible to explain the objective effect, which facilitates the legal classification of the committed act as a crime or crime, which allows for the tightening or leniency of the sanction provided for in criminal law.

If we denote the general class of these norm as CN, and the symbol ∩ denotes logical conjunction, the mutual relations presented in Fig 1 can be expressed in Formula (1), respectively:

$$CN = \{E, \ M, \ L, \ (E \cap M), \ (L \cap M), \ (E \cap L)(E \cap M \cap L)\} \qquad (1)$$

As moral norms will be the main point of reference in our experimental approach, we will now pay more attention to the subclasses from Formula (1) which contain only such norms.

We will define these kinds of moral norms as pure or context-free. The first such subclass, symbolized as $\dot{M}$, denotes the moral norms which belong only to the universe of moral norms but which do not belong to executive or legislative norms. As such, and where $x$ symbolizes any moral norm, and $\varepsilon$ is a predicator denoting *belonging to*, this subclass may be strictly defined as in Formula (2):

$$\dot{M} = \{x : x \in M \cap \{x \notin E \cap x \notin L\}\} \tag{2}$$

The second two subclasses are the conjunctions of moral norms with legislative norms: M∩L and moral norms with executive norms: E∩M. The appropriate formal explications can be viewed in Formulas (3) and (4), respectively:

$$M \cap L = \{x : x \in M \cap L\} \tag{3}$$

$$M \cap E = \{x : x \in M \cap E\} \tag{4}$$

There is also a subclass of moral norms in Formula (1) which is a conjunction of the three norms, executive, moral, and legislative: E∩M∩L, which is expressed in Formula (5) as:

$$E \cap M \cap L = \{x : x \in E \cap M \cap L\} \tag{5}$$

The moral norms presented above will be employed in our experiment in the context of moral situation referred to as *ethical stories* or briefly, as *moralities*. This method has already been employed in psychological research on moral issues, e.g., by Piaget [26] and Kohlberg [10] to diagnose the moral development of children and youth.

It is worth reviewing the strong and weak aspects of using *ethical stories* in psychological research. We propose conducting a short SWOT analysis of *moralities* in empirical research to confront this method with personal experience or observations of real-life moral situations. The results of this analysis are collected in Table 1.

From the outcomes of SWOT analysis, we learn that ethical stories are more easily employed in nomothetic empirical analysis on moral situations. It must be stressed clearly, however, that the subject of our research will be the cognitive aspect of ethical evaluation related to the concept of *morally good performance in moralities* analysis. Cognitive schemas of ethical stories offer the keys to understanding the concept of internalized behavior.

## Cognitive schemas of ethical stories

Let us review the main concepts related to ethical stories. Indicators of moral context relate to analysis of cognitive schemas of ethical stories. One indicator is a *moral requirement* or *demand* to do something in a defined situation, which is usually expressed by employing the

**Table 1. The outcome of SWOT comparative analysis outcomes of direct observation of real-life moral situation and ethical stories (moralities).**

| Evaluation \ Methods | Personal experience or direct observation of a moral situation | Ethical stories or moralities |
|---|---|---|
| Strong sides of the method | 1. Acknowledges the rich context of a moral situation. | 1. Enables grasping *hic et nunc* the ethical essence of the moral situation. |
| | 2. Enables employing an idiographic analysis of ethical evaluations of human behavior. | 2. Enables employing nomothetic oriented research in ethical evaluation of human behavior. |
| Weak sides of the method | In many moral situations the moral motives and intentions are hidden to the external observer. | Abstracts from the entire context of the moral situation. |

functor of assertive judgement *should* that was systematically elaborated by Immanuel Kant as one of his three formulations of a *categorical imperative (kategorischer Imperativ)*, which is the voice of human conscience–expressed by Kant as: "Act only in accordance with that maxim through which you can at the same time will that it become a universal law" (Kant, 1966, 4:421) [27]. Without this indicator, none of the moral norms defined in Formulas (1)–(5), could exist at all. If we state that the symbols *CI* denote Categorical Imperative, we can conclude that the norm regulating one's behavior is moral. This moral condition can be expressed as a logical alternative connection (expressed by symbol U) of the particular subsets of moral norms defined in the appropriate Formulas (2)–(5) and visually presented in Fig 1 as the corresponding fragments of Venn diagram:

$$x \in CI$$
$$\Rightarrow \{[(x \in M) \cap (x \notin E) \cap (x \notin L)]U[(x \in M \cap L)]U[(x \in M \cap E)]U[(x \in E \cap M \cap L)]\} \quad (6)$$

Another indicator of morality is here the object of moral regulation, which is understood as a content or domain regulated by the particular moral norms indicated in (6). Morally good performance assumes a specific object within which a particular moral norm has been actualized. Good examples could be the ethical codes of various professions. It is precisely stated in such codes that particular content-oriented professional activities are extracted from the remaining professional activities to be regulated by moral norms.

Also belonging to the indicators of morality are what causes a moral norm to become the given norm of the content of a the concrete ethical story, that is, what specifically distinguishes the given norm (specified in Formula (6)) from any other outside normative moral activity of a human being (as regulated separately only by legislative–L, or executive–E norm, as seen in Fig 1 and in Formula (1)).

After reviewing the literature on indicators of morality [10, 26, 28], Grác [1, 12] decided to apply the following six indicators for the experimental design: intention–I; concordance–C; application–A; outcome–E; knowledge–K; and realization–R. Realizing that moral reasoning related to ethical evaluation of human behavior in real-life situations is definitely a complicated mental process, we have decided to restructure the above indicators into complementary experimental units, in the two cognitive schemes expressed formally as:

$$IAO = I \cap A \cap O \quad (7)$$

$$KCR = K \cap C \cap R \quad (8)$$

The issue is: what values may the indicators in the Formulas (7) and (8) take? It is not a problem to apply the mathematical symbols (+) and (-) to denote, by analogy, an evaluation of human behavior in the ethical story as morally good or bad, respectively. However, these two bipolar values are not sufficient for moral evaluation of ethical stories, because in these stories we can meet with human behavior which, for various reasons, is beyond moral evaluation. That is why we have introduced a third neutral value that may be assigned to each indicator employed in our experiment. The symbol Ø taken from Boolean algebra is applied here. This value will denote that in terms of the specific indicator, the analyzed human performance is outside or beyond the specific aspect of moral evaluation. Moreover, it has to be emphasized here that in the cognitive schemas constructed in this experiment, the denotation of each indicator of morality is precisely defined not only by marking it as (+), (-) or Ø, but also by giving a particular interpretation to it in respect to the specific moral dimensions in these schemas. These symbols will be called antecedent marks in our analysis. We term these specific interpretations as *definiens*, which will be precisely defined for the particular schemas (see Tables 2

**Table 2. The indicators of morality aspects and their definiens, i.e., criteria for their ethical assessment in the IAO schemas.**

| Indicators of morality aspects | Criteria for their ethical assessment (definiens) | | |
|---|---|---|---|
| **I**NTENTION–the behavior intention, incentive and the moral norm are: | (+) | (-) | (∅) |
| | in accord | in discord | neither |
| **A**PPLICATION–the form applied in this particular case in relation to the moral norm: | (+) | (-) | (∅) |
| | corresponds | does not correspond | neither |
| **O**UTCOME–consequence accompanying, possible outcome is: | (+) | (-) | (∅) |
| | positive | negative | neither |

and 3). Next, we present the cognitive schemas of IAO and KCR as the matrices for this experiment on moral reasoning.

## IAO cognitive schemas

Indicators for the moral assessment and their definiens, i.e., criteria for their evaluation are defined in Table 2.

These indicators are specific conditions under which any moral norm works in a certain life situation. The criteria are definientia (sing. *definiens*) of the moral norm effect, expressing the fact that each of the perpendicularly identical symbols (+—Ø) has a different notional content–depending on its combination with the indicators.

The first indicator of the IAO schema is Intention–I. It is understood as a motive or intent of the individual's ethical behavior. As an individual cannot be forced to be moral, the axiological principle in ethics assumes his or her deliberation and free decision-making in the moral norm interactions. For this reason, this indicator is regarded as exceptionally important. But the indicator of the person's motive or intention cannot be absolutized since the individual's conscious relationship to a moral norm is an inevitable but not sufficient precondition for an ethics-based assessment of the person's behavior. How the concrete behavior will be evaluated from the perspective of intention and motivation depends on the criterion definientia belonging to this indicator. If, for example, the motive or intention of the behavioral event is in accord with the moral norm, then the result of the partial evaluation of behavior is positive (+). If it is in discord, the result is negative (-). The symbol Ø (*nullus*) signals that neither of the previous assessments are valid. Usually, this refers to activities which, due to their content, are regarded as beyond moral, but also to those where the motive of behavior is regulated by a moral norm, but the person is not aware of it.

Application–A, the second indicator of the IAR scheme, answers the question of whether the intention which is moral is or is not obtained by moral means. If the means of the intent's actualization corresponds with the moral norm, the result is positive (+). If the person tried to achieve the intended good by unethical means, their behavior is evaluated negatively (-). This means that the application indicator always assumes re-activation of the original intent and thus there is also the possibility of a greater or smaller change in the application process; consequently, this resultant motive may, but also may not, be in accordance with the original motive. The fact that every application implies not only motivation but also its actualization gives this indicator, according to the IAO scheme, a crucial position in the evaluation of the individual's ethical behavior.

The third indicator of the IAO scheme is Outcome–O, as a possible or an accompanying outcome. The indicator in question is related to the assessed behavioral event. The concept of relation itself can be understood as causal or non-causal. It could be causal in the case where the intention indicator was the cause of the actualized intention and the consequence indicator

was the impact of it. Since the function of the actualized intention was already adopted in the IAO scheme by the intentional indicator (as mentioned above), it is not possible–in the framework of the same scheme–to assign it simultaneously to the third indicator. Therefore, the Outcome indicator can be understood in the given scheme exclusively as non-causal, i.e. as a consequential circumstance which is related to cause and effect, but which cannot be identified with it. From the above-mentioned analysis, it follows that the assessment of the person's behavior by means of three signs (+—Ø) applies only to the evaluation of the accompanying consequence *per se* and does not influence the overall evaluation of the individual's behavior in the given moral situation.

Synthetically, the ethical evaluation processes in a moral situation within the cognitive scheme of IAO could be formulated as follows (9):

$$\Pi_{I,A,O} \sum_{\{x:x \in \{(+),(-),\emptyset\}\}}$$
$$= \{[I = (+) \cup I = (-) \cup I = (\emptyset)] \cap [A = (+) \cup A(-) \cup A = (\emptyset)] \cap [O = (+) \cup O = (-) \cup O = (\emptyset)]\}(9)$$

, where: the symbols I,A and O–denote the morality indicators in the experimental cognitive schema IAO, i.e. Intention, Application, Outcome, respectively; (+),(-) and (Ø)–are the possible three values which can take each of the morality indicators in modeling phase of the experiment; U–denotes logical alternative connection of these values; and ∩–denotes logical conjunction of the three compounds which are the morality indicators in cognitive scheme of IAO.

## KCR cognitive schemas

Indicators of moral assessment related to the cognitive schemas KCR and the criteria for their evaluation are presented (so called *definientia*) in Table 3.

The first indicator of morality in the KCR schema is Knowledge about moral norms, which regulates human behavior in a given situation. This is really moral knowledge about the specific norm, which can exist as a socially accepted norm within public opinion, regulating interpersonal relations. However, the acting person may or may not have knowledge about this opinion. Thus, the cognition of the moral norm can be very differentiated. The norm can exist as a socially accepted ethical obligation which a concrete individual might not know about. There could also be cases where an individual has a positive attitude toward the defined requirement and accepts it as his or her own moral norm, independently of whether such norm objectively exists as a social standard or not. The symbol Ø can be used to value this indicator, if is really unclear as to whether to assign a value (+) or (-).

The second indicator in KCR is Concordance (C) of the person interacting in a given moral situation. Concordance means the actor's personal attitude to the particular moral norm regulating human performance in this situation. According to Grác (2008, 2015), using the signs: (+) or (-) in valuing the indicator C is relatively univocal, but some problems concerning

**Table 3. The indicators of morality aspects and their definiens, i.e. criteria for their ethical assessment in KCR schemas.**

| Indicators of morality aspects | Criteria for their ethical assessment (definiens) | | |
|---|---|---|---|
| **K**nowledge–as a cognitive component to the specific moral norm | (+) | (-) | (∅) |
| | exists | doesn't exist | neither |
| **C**oncordance–acceptance of the particular moral norm | (+) | (-) | (∅) |
| | is accepted | is not accepted | neither |
| **R**ealizing–the moral norm has actually been implemented | (+) | (-) | (∅) |
| | fulfilled | not fulfilled | neither |

understanding may require the sign Ø. This sign should be employed if the evaluating person has neither a positive nor negative attitude towards the given moral norm or when this person does not have his or her own understanding of the moral requirements related to the norm in question.

The last indicator Realizing–R refers to whether and how the requirements of the moral norm regulating human behavior are fulfilled. A positive (+) value of this indicator signifies that implementation of the moral norm is fulfilled. This indicator receives a negative value if implementation of the moral norm is not fulfilled in a situation where it is supposed to be. Ethical evaluation faces greater difficulty when considering whether to assign the value Ø. This case is particularly difficult to assign due to the fact that there is no analogy between a neutral attitude towards a moral norm (as in the case of indicator C above) and the neutrality of acting in a concrete situation, when we are dealing with human performance.

In concluding this stage of our analysis, we can state that ethical evaluation processes in a moral-content-situation within the cognitive scheme of KCR can be formulated as follows (10):

$$\{\prod_{\mathbf{K},\mathbf{R},\mathbf{C}}\sum_{\{\mathbf{x}:\mathbf{x}\in\{(+),(-),\emptyset\}\}} = \{[\mathbf{K} = (+) \cup \mathbf{K} = (-) \cup \mathbf{K} = \emptyset] \cap [\mathbf{R} = (+) \cup \mathbf{R}(-) \cup \mathbf{R} = \emptyset]$$

$$\cap [\mathbf{C} = (+) \cup \mathbf{C} = (-) \cup \mathbf{C} = \emptyset]\} \tag{10}$$

, where: the symbols K,C and R–denote the morality indicators in the experimental cognitive schema KCR, i.e. Knowledge, Concordance, Realizing, respectively; (+),(-) and (Ø)–are the possible three values which can take each of the morality indicators in modeling phase of the experiment; U–denotes logical alternative connection of these values; and ∩–denotes logical conjunction of the three compounds which are the morality indicators in the cognitive schema of KCR.

## Control of independent variables in evaluating cognitive schemas of ethical story experiments

The main methodological requirement for any experimental condition is to control the independent variables which make an impact on the dependent variable. In our experiment on moral thinking, the main dependent variable to be explained is the ethical evaluation of human behavior in terms of moral values. It is assumed that moral reasoning is a structured mental process like any other human thinking structured towards problem solving. Since John Dewey's *How we think* [13], psychologists working on problem solving, have distinguished the phases of thinking which lead human thinking to reach the goal of solving a problem. Similar thinking takes place in a real-life moral situation where a human being considers whether and which action should be taken in order to behave in a morally acceptable way, in accordance with his or her conscience–based on a moral norm.

Our experiment will be based on the evaluation of ethical stories (moralities), using the defined structured indicators of morality, called cognitive schemas of the ethical-dilemma-stories, described above as IAO and KCR.

The first novelty of this experimental approach is the two-phase design: (1) modeling of ethic-dilemma stories based on logically structured cognitive schemas, and (2) overall ethical evaluation of the modeled morality schemas.

Assessing the effect of any moral norm in conditions of an experiment situation, given by the schemas IAO and KCR, is one methodological challenge; another is grasping these conditions in a person's actual behavior. This can be done by means of modeling written stories which present real-life moral situations. These stories enable the observer to examine

intrinsically experienced behavior, e.g., their motivation or intention, which is usually hidden from a second party's retrospection.

We use the term *modeling* for the creation of moral dilemma stories. This involves the assignment of arbitrary content of actually displayed behavior to the respective value indicators of the schema, so that this allocation may accurately correspond to the criterion definientia of the indicators (I,A,O,K,C,R) of morality. Educational psychology students were the subjects of this experiment. They were asked to create model ethical stories concerning moral situations of young students of their age, according to carefully-prepared instructions informing them how to do this task.

## Control of independent variables in the experiment

Three question arose in the definition of *modeling*. The first of these is formulated as follows: how many *independent variables*, that is, different configurations of signs, are offered for story modeling by the schemas IAO and KCR, respectively?

In order to answer this question for IAO schemas we have to strictly consider the three indicators of morality where each of them, formally speaking, may take three possible values: (+), (-), (Ø). Thus, to calculate all configurations of each indicator combinatorially, where generally its number is *n*, where the number of values that each of the indicators can take may be generally symbolized by *r*, we could apply the formula for variations with repetition $n^r$. This means that variations with *n* elements of the $r^{th}$ class are all possible configurations of *n* indicators of morality and the *r* values which each indicator can take. In our case, where the number of indicators *n* = 3 in the IAO schemas, and the number of values which each of them can take is r = 3, the number of all configurations is simply attained using Formula (11) as follows:

$$\{N(IAO) : N(IM) = 3 \cap N(VIM) = 3\} \Rightarrow \{N(IAO) = 3^3 = 27\} \tag{11}$$

where:

N(IAO)–the total number of independent variable configurations related to the indicators of morality and their values in the IAO schemas;

N(IM)–the number of indicators of morality in the IAO schemas;

N(VIM)–the number of values that can be taken by each of the indicators of morality in the IAO schemas.

And, similarly, for the KCR schemas, where the number of indicators is also *n* = 3, and the number of values which each of them can take is *r* = 3 as well, we can reach the number of all configurations simply by using Formula (12) being a replication of Formula (11) as follows:

$$\{N(KCR) : N(IM) = 3 \cap N(VIM) = 3\} \Rightarrow \{N(KCR) = 3^3 = 27\} \tag{12}$$

where:

N(KCR)–the total number of independent variable configurations related with the indicators of morality and their values in the KCR schemas;

N(IM) and N(VIM)–are as in Formula (11).

## The elementary experimental fields of moral evaluation: Ethical stories

The second question is whether the same story content, which is modeled according to certain specific criteria of evaluation, can be *modified* in accord with further randomly applied evaluation criteria of the IAO or KCR schemas. In an effort to answer this question, we can alternately replace each indicator of morality by one of the criterion values: (+), (-), (Ø). If we take as an example the description of three 18-years-old youths who solved the problem of moving to a new place to live in different ways, we obtain the IAO and KCR schema matrices with

**Table 4. Matrix of IAO schemas illustrating the elementary experimental fields of moral evaluation in ethical stories.**

| columns / rows | $c = 1$ | $c = 2$ | $c = 3$ |
|---|---|---|---|
| | I (+) | A (−) | O (∅) |
| $i = 1$ | John found a new flat to create better conditions for his siblings to study. | He did it regardless of the disapproval of his mother, who felt almost mortally hurt by his decision. | After having moved to his new flat John found out that the signal for his cell phone was the same quality as in the previous location. |
| | I (−) | A (∅) | O (+) |
| $i = 2$ | Paul found a new flat to have easier access to drug dealers. | While moving to the new flat, he decided to travel by train instead by bus. | At the new location he met a neighbor who began to exert a positive influence on him. |
| | I (∅) | A (+) | O (−) |
| $i = 3$ | Peter found a new flat because he wanted to live alone. | By mutual consent with his parents he left their home. | While moving to the new flat he slipped and broke his leg. |

nine boxes as a part of 27 total number of combinatorial boxes, accounted separately for the IAO and KCR cognitive schemas, as presented in Formulas (11) and (12), respectively. For the schema matrixed in our experiment the total number of rows $i = 3$, and the number of columns is $c = 3$. If any line is denoted as $i$ and any column as $j$, then any field in these matrices can be expressed as an experimental field $EF_{ij}$ of partial moral evaluation of human performance in ethical story. An example of such $EF_{ij}$ are shown in Table 4 for the IAO schemas.

As in our case, the IAO scheme is a square matrix, the values in the first line can be written symbolically: $i_1 c_1 = I (+)$, $i_1 c_2 = A (—)$, $i_1 c_3 = O (∅)$ and at the same time they can be checked by reading them verbally in the first row of the chart. The square matrix enables us to read the same story plot variably, i.e., according to an arbitrary configuration of boxes in the lines, without losing the inner logic (meaning). For example, if we select only positive criteria, then the story plot can be expressed in the following notation: $i_1 c_1 = I (+)$, $i_3 c_2 = A (+)$, $i_2 c_3 = O (+)$. When we read it in the chart in this transcript, we find out that the story plot did not lose its inner coherence. It is possible to symbolically record only the minus or only the null configuration in the same way. However, any random two-element combinations are also possible, e.g., combination of two negative and one positive value. In that case, the notation is as follows: $i_2 c_1 = I (—)$, $i_1 c_2 = A (—)$, $i_2 c_3 = O (+)$.

Based on the above, the IAO schemas make it possible not only to create stories differing in their content, but they also offer the opportunity to modify the already-modeled stories.

In a similar way, we will now give an example of experimental fields $EF_{ij}$ of the schemas KCR shown in the square matrix presented in Table 5.

**Table 5. Matrix of the KCR schemas illustrating the elementary experimental fields ($EF_{ij}$) of moral evaluation in ethical stories.**

| Columns / lines | $c = 1$ | $c = 2$ | $c = 3$ |
|---|---|---|---|
| | K (+) | C (−) | R (∅) |
| $i = 1$ | Igor knew that cruelty to animals is immoral. | He was not opposed to ill-treatment of animals. | However, he has never even been in a situation where he watched cruelty inflicted on animals by others. |
| | K (−) | C (∅) | R (+) |
| $i = 2$ | Andrew did not know that cruelty to animals is immoral. | In view of the abuse of animals, he was unclear and of a contradictory position. | Yesterday, when his colleagues mistreated a dog, he protected it. |
| | K (∅) | C (+) | R (−) |
| $i = 3$ | Emil was not sure if cruelty to animals is moral or immoral. | But personally he did not agree with the abuse of animals. | However, yesterday, when colleagues mistreated a dog, he was also cruel to it. |

From Table 5 we can see that KCR schemas, similar to IAO schemas, also have a variable structure of stimuli constituted by the content of combinatorial sequences of elementary experimental fields ($EF_{ij}$) at a level equal to IAO. These experimental fields are indicators of morality of each individual ethical story presented in both the IAO and KCR schemas. The experimental control of independent variables in this experiment deals with combinatorial control of the particular scheme in terms of the formal values (+), (-) and $\emptyset$ in IAO and KCR, respectively. Moreover, in order to avoid contradictions between the concrete schemas, on the one hand, and to maintain individual and personal coherence of the ethical story content, on the other, the subjects were the heroes of each of the 27 IAO schemas and 27 KCR schemas, where each schema actor was assigned a different name in the modeled ethical stories.

## Experimental design

In our experiment the stimuli are the individual schemas, which consist of three indicators of morality ordered in the two following sequences: IAO and KCR, where each of the indicators takes only one of three values in its matrix of sequences (see Tables 4 and 5). The valuated indicators of morality constitute a concrete cognitive schema as a complex sequential stimulus which consists of three particular experimental fields as the objects of ethical individual story modeling in the first phase of the experiment.

Thus, each individual IAO schema can be defined as a well-ordered set of three:

$$Si(\text{IAO}) \overset{\text{def}}{=} \langle I, A, O \rangle, \tag{13}$$

where: $S_i(\text{IAO})$—any of the $n$ ($n = 27$ in our experiment) IAO schemas; I = {(+),(-),($\emptyset$)}; A = {(+),(),($\emptyset$)}; O = {(+),(-),($\emptyset$)}–denote the sets of values which can take particular indicators of morality in the IAO schemas. This indicates that the order of the indicators constituting the IAO schema should not be changed.

In a similar way, each individual KCR schema can be defined as a well-ordered set of three:

$$Si(KCR) \overset{\text{def}}{=} \langle K, C, R \rangle, \tag{14}$$

where: $S_i(KCR)$—any of n (n = 27 in our experiment) KCR schemas; K{(+),(-),($\emptyset$)}; C = {(+), (-),($\emptyset$)}; R = {(+),(-),($\emptyset$)}–denote the sets of values which can take the particular indicators of morality of the KCR schemas.

The methodological analysis conducted allows us to formulate an experimental design to present cognitive schemas for ethical story modeling by educational psychology students.

**Table 6. The experimental design: The structure of valuation of indicators in particular IAO and KCR cognitive schemas for experimental modeling of ethical stories by educational psychology students.**

| IAO schemas | Modeling formal patterns | KCR schemas | IAO schemas | Modeling formal patterns | KCR schemas | IAO schemas | Modeling formal patterns | KCR schemas |
|---|---|---|---|---|---|---|---|---|
| 1 | (+ + +) | 28 | 10 | ($\emptyset$- -) | 37 | 19 | (- $\emptyset$ +) | 46 |
| 2 | (+ - +) | 29 | 11 | (- - $\emptyset$) | 38 | 20 | (+ - $\emptyset$) | 47 |
| 3 | (- + +) | 30 | 12 | ($\emptyset\emptyset\emptyset$) | 39 | 21 | (+ $\emptyset$ -) | 48 |
| 4 | (+ + -) | 31 | 13 | (- - +) | 40 | 22 | ($\emptyset$ - $\emptyset$) | 49 |
| 5 | (+ $\emptyset$ +) | 32 | 14 | (- + -) | 41 | 23 | ($\emptyset\emptyset$ -) | 50 |
| 6 | ($\emptyset$ + +) | 33 | 15 | (+ - -) | 42 | 24 | (- $\emptyset\emptyset$) | 51 |
| 7 | (+ + $\emptyset$) | 34 | 16 | ($\emptyset$ + -) | 43 | 25 | ($\emptyset\emptyset$ +) | 52 |
| 8 | (- - -) | 35 | 17 | (+ - $\emptyset$) | 44 | 26 | ($\emptyset$ + $\emptyset$) | 53 |
| 9 | (- $\emptyset$ -) | 36 | 18 | ($\emptyset$ - +) | 45 | 27 | (+ $\emptyset\emptyset$) | 54 |

These ethical stories involve moral issues from the students' own community. This design is included in Table 6 as a matrix of formal schemas numbered from 1 to 54 and presented in parallel columns in such a way that numbers for schemas IAO and KCR are located on either side of the formal valuation structures of the indicators of morality.

The order number of formal valuations of the IAO schema is always on the left side of the appropriate column for modeling formal patterns in Table 6 and the order number for the KCR schema is on the right side of this column, respectively. The signs of moral evaluation should be read from left to right both for IAO and KCR schemas.

From Table 6 we learn that both cognitive schema enable the planned experiment on moral thinking to control for variability of stimuli structure via evaluated indicators of morality of human performance modeled in experimental conditions during the students' classes.

## Subjects, experiment–(a) modeling instructions and procedure, and (b) overall moral evaluation of ethical stories

**Subject Group 1.** Subject Group 1 were 134 third-year educational psychology students of a modal age of 20 years, who participated in both parts of our experiment, i.e., in the modeling of ethical stories, and in the overall evaluating of these stories modeled by themselves. Their participation in this experiment came as part of a regular course on educational psychology at the Faculty of Philosophy of Trnava University in Trnava (Slovakia) over the years 1998-2000. 80% of the subjects were female students and 20% were males. After participating in theoretical lectures on cognitive schemas of ethical stories and evaluation of related indicators of morality, the students of the seminar on educational psychology were presented with the following task in accordance with these short instructions:

**Experiment instructions and procedure for Subject Group 1.** As you have learned today from the lecture on education psychology, each schema of an ethical story has its own order number in our experimental matrix and a concrete valuation of its three indicators of morality. Now you will receive data about two IAO schemas and two KCR schemas. Your task will be to creatively model the thinking and performance of a 17-year-old individual, different in each of the projected stories, in accordance with the given values of the individual indicators of morality according to the patterns of potential schemas, the order numbers of which will be generated by random number generator.

After this instruction each individual subject was given the *Ethical Story Modeling Experiment Report Form* (see the form with this protocol in the S1 Appendix) containing four order numbers associated with indicator values for potential ethical stories generated from Table 6 by a random number generator. The subject was instructed about where and what the modeling schema of the concrete story should be in accordance to the experimental design, and that the hero of each story should be an individual person known by his or her name.

Once the modeling phase of the experiment was completed, in the next phase the experimenter asked the same subject to conduct an overall evaluation of each of the constructed ethical stories in his or her protocol as a whole by assigning a global symbol in square brackets [] to denote their overall assessment of moral values of human performance as: morally good [+], morally bad [–] or morally neutral [∅] which are now called the concluding symbols.

The experimental modeling procedure and overall moral evaluation of ethical stories took place in a seminar room where the subjects could model the educational stories in quiet and comfortable conditions. They conducted their experimental modeling and overall moral evaluation individually, sitting separately at seminar tables in small groups. The average time for modeling was about 45 minutes. After completing the story modeling and conducting the overall evaluation of each of the constructed ethical stories, the students gave the completed

*Protocol for Experimental Modeling of Ethical Stories* and their completed *Moral Evaluation* back to the experimenter (see: S1 Appendix).

**Subject Group 2.** Taking into account that the studied group of students (Subject Group 1) participated in the entire experiment, i.e. in modeling and then in evaluating the entire ethical story, we can state that they had a fairly high level of ethical knowledge, particularly on overall evaluation of ethical stories, gained mainly from lectures on the psychology of morality. This, however, completely rules out the possibility of using the research results from the second part of the experiment as representative for the overall evaluation of the ethical stories constructed in our experiment by the young students.

Therefore, the authors of this study decided to select two other group of young people applying to study psychology for the second part of the experiment, i.e. for overall evaluation of the ethical stories (modeled in the first part of the experiment) as the subjects who are just after their maturation and participate in the entrance exams for psychology at the University of Trnava. These subjects are called Subject Group 2. Their number is 450 (N = 450).

## Results

The research findings from this moral reasoning experiment are presented according to the two phases of this experiment: (1) the results of modeling ethical stories, and (2) the results of the overall moral evaluation of the modeled ethical stories (see: the example of the *Protocol of Experimental Modeling of Ethical Stories* and its completed *Moral Evaluation* back to the experimenter (see: S2 Appendix).

### The modeling phase

The total number of protocols received with completed modeling was 536. This means that the main result of our findings from the modeling phase of our experiment on moral reasoning is a systematized empirical data base of ethical stories, which is a kind of empirical description of characteristics of denotative meaning of the indicators of morality understood in their positive, negative, or neutral meaning in a context of cognitive schemas of ethical stories. This base is systematized in accordance with the order numbers of the 54 IAO and KCR cognitive schemas, sequenced and presented as a matrix in Table 6. One way of presenting this data base to list the modal examples of modeling created on the base of this matrix. The full list of modal models can be found in the original Slovakian version of the report by Grác (2008, pp 119–125) and the Polish version by Grác (2015, pp.140-148).

This data base can be a source for qualitative analysis on the structuring of moral reasoning using indicators of morality. One of the opportunities for research using this data base points is to use the schemas for research on overall assessment of the moral value of the ethical stories. This is what was done in the second phase of this research project.

### The moral evaluation phase

We will now show what kind of cognitive strategies the subjects developed in their overall moral evaluations, based on analyzing the values of indicators of morality as premises in inferring the conclusions in the IAO and then KCR schemas.

### Strategies used to complete overall moral evaluations of the IAO schemas of ethical stories

The students' overall moral evaluations of the IAO ethical stories allowed us to identify three regularities which describe their moral reasoning in analyzing the values of the indicators of

**Table 7. Global moral evaluations of the IAO schemas in accordance with the principle of univocal values of intention and application indicators.**

| Order number of the schema | IAO schema | Overall moral evaluation | Order number of the schema | IAO schema | Overall moral evaluation |
|---|---|---|---|---|---|
| 1 | (+ + +) | [+] | 12 | (∅∅∅) | [∅] |
| 4 | (+ + -) | [+] | 13 | (- - +) | [−] |
| 7 | (+ + ∅) | [+] | 23 | (∅∅ -) | [∅] |
| 8 | (- - -) | [−] | 25 | (∅∅ +) | [∅] |
| 11 | (- - ∅) | [−] | | | |

morality in these stories. These regularities are also intuitively deducible from the syntactic structure of the indicators of morality contained in the structures of formal ethical story schemas used in our experiment. These syntactic regularities are referred to as the model strategies of overall evaluations of ethical stories.

## The moral strategies of overall evaluations of IAO ethical stories

The first regularity can be called the *principle of univocal values of intention and application indicators*. The matrix which indicates the overall moral evaluations which fit this regularity is shown in Table 7.

The analysis of Table 7 allows us to formulate *a principle of univocal values of intention and application indicators* as follows:

$$(I = A) \Rightarrow (ES = I = A), \tag{15}$$

where: ES denotes the overall moral evaluation of ethical story; I and A denote the appropriate signs of moral values of the indicators in IAO schemas.

The second regularity of the overall evaluation of the IAO schemas of ethical stories is called the *principle of preferring application (A) indicator over intention indicator (I)*. Table 8 shows the matrix indicating the overall moral evaluations which fit this principle.

From Table 8 we can learn that *the principle of preferring application (A) indicator over intention indicator (I)* may be formally expressed as:

$$(I \neq A) \Rightarrow (ES = A), \tag{16}$$

where the symbols denote the same as in (15).

The third regularity says that if the intention indicator is non-zero, and the sign of the application indicator is zero, the concluded overall moral evaluation of the ethical story is formally expressed by the sign of the intention indicator, independently of the sign of the outcome indicator. This regularity is called as the *principle of preferring the non-zero intention indicator (I) over the zero application indicator (A)*. The designates of this principle are shown in the matrix presented in Table 9.

**Table 8. Overall moral evaluations of IAO schemas in accordance with the principle of preferring application (A) indicator over intention indicator (I).**

| Order number of the schema | IAO schema | Overall moral evaluation | Order number of the schema | IAO schema | Overall moral evaluation |
|---|---|---|---|---|---|
| 2 | (+ - +) | [−] | 16 | (∅ + -) | [+] |
| 3 | (- + +) | [+] | 17 | (- + ∅) | [+] |
| 6 | (∅ + +) | [+] | 18 | (∅ - +) | [−] |
| 10 | (∅ - -) | [−] | 20 | (+—∅) | [−] |
| 14 | (- + -) | [+] | 22 | (∅ - ∅) | [−] |
| 15 | (+ - -) | [−] | 26 | (∅ + ∅) | [+] |

**Table 9. Overall moral evaluations of IAO schemas in accordance with the principle of preferring the non-zero intention indicator (I) over zero application (A) indicator.**

| Order number of the schema | IAO schema | Concluded moral evaluation | Order number of the schema | IAO schema | Concluded moral evaluation |
|---|---|---|---|---|---|
| 5 | (+∅ +) | [+] | 21 | (+∅ -) | [+] |
| 9 | (-∅ -) | [−] | 24 | (-∅∅) | [−] |
| 19 | (-∅ +) | [−] | 27 | (+∅∅) | [+] |

Analysis of the patterns of indicator signs allows us to formulate in a formal way the principle of preferring the non-zero intention indicator (I) over the zero application indicator A expressed as:

$$[(I \neq \emptyset) \cap (\mathbf{A} = \emptyset)] \Rightarrow (ES = I). \tag{17}$$

## Representativeness of the syntactic models in overall evaluations of IAO ethical stories

At what level are, the above-presented models of overall evaluations of IAO ethical stories, descriptive for the subjects? Overall evaluations of IAO ethical stories extracted from the empirical data base were carried out, having been obtained in the modeling part of the experiment and prepared in the ZEU-IAO Test for further research. The participants of this part of the experiment (N = 450) were high school graduates who took their entrance exams for psychology at the University of Trnava (Slovakia). In 93.75% of the IAO ethical stories schemas, the modal evaluation of the subjects turned out to be consistent with the model strategy of logic-formal syntax. The percentages of these strategies ranged from 28.70% to 98.00% [1].

These results were consistent with a study carried out three years later with another group of high school graduates (N = 355) participating in the entrance exams for psychology at the same university. They were tested with the same ZEU-IAO test as the group of their colleagues earlier. Both study groups had no contact with each other in the interim. The second study showed that for 93.75% of the IAO ethical stories schemas, the modal evaluation of the subjects turned out to be consistent with the model strategy of logic-formal syntax indicated in Tables 7–9 and in Formulas (15)–(17), respectively. This means that the strategy of integrating the morality indicators of moral stories into IAO schemes resulted in highly descriptive models for the processes of moral evaluation of these stories. The percent of subjects employing these strategies in the experimental ethical stories ranged from 32.00% to 98.30% [1].

The relatively high sample of subjects in two consecutive studies (N = 450 and N = 355) provides a strong indication that modal percentages of moral evaluations are a good fit for the model strategy of integrating moral indicators into the decision-making situations of ethical evaluations in our experiment. However, the fit with the model evaluation is strongly conditioned by the context of the moral situation itself. In the ethical stories of the IAO scheme, differences in moral conclusion congruence are as high as 70%.

## Strategies used to complete overall moral evaluations of the KCR schemas of ethical stories

The first main finding concluding the moral evaluation of the ethical stories within the KCR schemas is that only concordance (C) and realization (R) indicators of morality played a decisive role in the overall moral evaluations of the stories. Familiarity with the heroes of the stories of moral norms (K) appeared only to play a contextual role in the ethical stories, with no impact on completing the moral evaluation process.

The students' assessment data allowed us to identify the following four strategies in concluding the moral evaluations of the KCR schemas of ethical stories:

1. If the sign of the concordance indicator (C) is univocal with the sign of the realization indicator (R), the sign of the concluded moral evaluation of the ethical story is equal to the signs of both indicators in the premises of this conclusion, which can be formally expressed as:

$$(C = R) \Rightarrow (ES = C = R), \tag{18}$$

where: ES–denotes the same as in the above formulas, and C, R denote the symbols of the KCR schemas.

2. If the sign of the concordance indicator (C) contradicts the sign of realization indicator (R), the overall moral evaluation is expressed with the sign $\emptyset$. As a formal expression it is stated as:

$$(I \neq R) \Rightarrow (ES = \emptyset) \tag{19}$$

3. If the sign of the concordance indicator (C) is neutral ($\emptyset$),the overall concluded moral evaluation of the ethical story is neutral as well, which one can formulate as:

$$(C = \emptyset) \Rightarrow (ES = \emptyset) \tag{20}$$

4. If the sign of the realization indicator (R) is neutral ($\emptyset$),the overall moral evaluation of the ethical story takes its value from the non-zero evaluation of the concordance indicator (C). In a formal way this regularity may be expressed as:

$$(R = \emptyset) \Rightarrow (ES = C) \tag{21}$$

## Representativeness of the syntactic models in overall evaluations of KCR ethical stories

In this section we indicate to what extent the integration stretches of morality indicators in the overall evaluation of ethical stories according to the KCR scheme (formulated in Table 10 and Formulas (18)–(21)) constitute descriptive models for students participating in the experimental evaluation of these stories. To do so, we use the test results of a group of high school graduates (N = 450) who participated simultaneously in the test (both the ZEU-IAO Test and the ZEU-KCR Test). Based on the general data for the degree of compliance between the empirical distribution of moral valuation of the respondents for the tested ethical stories and the indications of the models discussed above, it can be said that in 10 test tasks out of all 16 tasks displayed for the KCR scheme, model strategies turned out to be descriptive for the evaluation of the studied group of high school graduates. This means that in 62.50% of evaluation tasks, the modal number of respondents in 10 out of 16 groups of tasks, was consistent with the logical-formal syntax model. A more detailed presentation of this research situation is indicated in Table 10.

The data from Table 10 allow us to more accurately conclude that the modal percentages of model evaluations are quite varied with respect to individual moral stories exposed in the

**Table 10. Percentage indicators of modal evaluations of moral stories KCR for the choices of respondents in relation to model indications.**

| The order of the ethical story in ZEU-KCR | Values of morality indicators in the | Overall evaluation of the moral story | Modal percentages of model evaluations* | Modal percentages of non-model evaluations |
|---|---|---|---|---|
| | **K C R** | | | |
| 01 | (+) (+) (+) | [+] | **79.4%** | |
| 02 | (+) (-) (-) | [–] | **49.2%** | |
| 03 | (-) (∅) (-) | [∅] | **64.0%** | |
| 04 | (+) (-) (+) | [∅] | 36.4% | 58.2% [–] |
| 05 | (-) (∅) (+) | [∅] | **50.9%** | |
| 06 | (-) (∅) (+) | [∅] | **70.2%** | |
| 07 | (+) (+) (+) | [+] | **89.0%** | |
| 08 | (+) (-) (-) | [–] | **89.2%** | |
| 09 | (-) (∅) (-) | [∅] | **68.3%** | |
| 10 | (+) (-) (+) | [∅] | 27.0% | 68.6% [–] |
| 11 | (+) (-) (-) | [–] | **70.2%** | |
| 12 | (+) (+) (-) | [∅] | 15.8% | 82.4% [–] |
| 13 | (+) (-) (+) | [∅] | 44.7% | 49.4% [–] |
| 14 | (+) (-) (-) | [–] | **86.1%** | |
| 15 | (+) (-) (+) | [∅] | 36.5% | 61.5% [+] |
| 16 | (+) (+) (-) | [∅] | 19.8% | 75.2% [–] |

* The bolded percentages in the column fourth of Table 10 denote that these modal evaluations by the subjects fit in the defined ethical stories, the model strategies related to these stories.

experiment: from 49.2% to 89.2% of the respondents' modal moral evaluations. The data clearly indicate that students' moral evaluations of ethical stories under the KCR scheme are significantly more consistent with the logic-syntactic models of moral evaluation of these stories. The difference between the most difficult and easiest evaluations here is 40%, whereas in the IAO scheme it was 70%.

## Conclusions and final remarks

In this section we will attempt to respond, first of all, to the question of which mental processes are involved in both the modeling phase of our experiment and the overall evaluation of the ethical story. Grác [1, 12] assumes that in modeling ethical stories, semantic operations are involved to create thematic-oriented structures in accordance with the sequences determined by the formal signs of the indicators of morality which constitute the cognitive schemas of potential stories (moralities). These semantic mental operations must be focused both on securing particular sequences for the appropriate moral valuation required by the cognitive schemas obtained in the experiment, and, at the same time, on maintaining the thematic coherence of the content of the whole ethical story.

We consider here the role in moral reasoning played by the particular indicators of morality known from the literature, such as motivation and intension [28–31]. In the overall evaluation phase of the experiment, moral reasoning required more syntactic and formal logic cognitive operations. Grác [1, 12] distinguishes in the structure of human mind the three processes that enabled the subjects to formulate the regularities in concluding the overall moral evaluation of ethical stories: *selection*, *differentiation* and *integration*. If we compare J. Dewey's [13] first five-stage structuring of the goal-oriented thinking or problem-solving process to the Gràc model, we can interpret Gràc's (2008) distinguished mental processes precisely as stages in the process of moral reasoning toward a goal which is the moral evaluation of a specific ethical story in an

experiment. Hence, when faced with the problem of moral evaluation of a particular ethical story, the subject of that decision undertakes the following three mental processes or phases: 1. the phase of selection processes; 2. the phase of differentiation processes; and 3. the phase of integration processes.

The *selection process* deals with highlighting the indicators of morality that do not influence moral evaluation of ethical stories, but only create a situational context in these stories. These indicators appeared to be: the outcome indicator (O) in the IAO schemas, and the knowledge indicator (K) in the KCR schemas. This means that these two indicators are not considered as premises in completing overall moral evaluation of ethical stories. The result of the *differentiation process* is in distinguishing among the indicators of morality which play a dominant role in competing the overall moral evaluation of moral stories, and those which are only connected with this evaluation. The dominant positions were taken by: the application indicator (A) in the IAO schemas and the concordance indicator (C) in the KCR schemas.

The main mental process related to conducting an overall moral evaluation is *integration*, which allows a generalizing value to be assigned for each group of three indicators in the IAO and KCR schemes. As a result of this process, seven regularities were identified that describe the generalized patterns of indicator sign in the premises and the inferred overall conclusion for each group of ethical stories. The relationship discovered between each of the particular groups of indicator signs as premises and the overall moral evaluations of the moralities constitutes a psychological connection for moral inference in our experiment on moral evaluations of ethical stories. Formally, this relationship can be expressed in the formal logic syllogism *modus ponendo ponens* as follows:

$$[(p \Rightarrow q) \cap p] \Rightarrow q, \tag{22}$$

where the symbols of inference connection ($p \Rightarrow q$) denote the particular regularities formulated in (17)–(21) as the first premise for conducting the moral evaluation, where $p$ denotes the conditional sentence concerning the particular group of indicators signs of moral valuation and $q$ denotes the result of the overall moral evaluation of ethical story.

In conclusion, we have described the novelty of the two-phase experiment conducted on moral reasoning, which included ethical stories modeling phase and an overall moral evaluation phase where the subjects were psychology students. The ethical stories modeling phase developed by the students gave them the chance to learn what moral reasoning about youth behavior is in an attractive way, and what role the indicators of morality of their experimentally modeled ethical stories play in overall moral assessment of human performance. This phase of the experiment enabled the subjects to take a large step towards the process of internalization of moral norms regulating human behavior in social situations. In addition, this process was of a multidimensional nature and a structured character, leading to the overall moral evaluation of human action.

Reflecting on our theoretical analyses and experimental studies of moral reasoning, we can draw the following conclusions: (a) moral reasoning is a multi-faceted psychosocial phenomenon, where its mutually linked factors constitute the moral existence of the individual human person; (b) there is a need to distinguish and differentiate the following components of moral reasoning processes for both theoretical and practical purposes: cognitive, volitional, emotional, motivational (intentional) and behavioral processes–which the literature refers to as indicators of morality; (c) psychology is still far from determining the external validity of its psychometric and experimental instruments for studying moral reasoning in natural experiment circumstances with a higher ecological validity.

The uniqueness of our experiment lies in the fact that the students of psychology themselves, properly prepared by the ethical story modeling program during the first stage of the experiment, called modeling, developed the content of complex matter story schemas for the second stage of the experiment, using the indicators of moral evaluation of their own youth social performance. On the other hand, new students who were just applying for psychology at the same university participated in the second stage of the experiment, and had not yet had any contact with their older colleagues who were the subjects in the first stage of the experiment. Thus, the contexts of ethical storytelling schemas, modeled by the students of psychology, received an overall moral evaluation by their slightly younger (a year or two) colleagues. In this way, a satisfactory ecological validity of the results of our experimental research on moral reasoning was achieved.

It is for these reasons, we may say that this experiment took place in closer proximity to real-life thinking processes than other research methods previously used in experimental psychology. Such a conclusion is justified both on the basis of the meta-analysis of the psychological literature from 1940–2020 on morality (see: [11]) as well as our own analysis of publications from the last three years (e.g. experiments made by Sudić & Ćirić, [21]; by Awad, [25]; and by Rhim, Lee & Lee, [22]) what was presented earlier in our paper. This means that we achieved significantly higher ecological validity in the presented experiment than has been achieved thus far in other experiments on moral reasoning. However, this does not mean that the level of ecological validity in our experiment is already totally satisfying.

Let us now try to indicate the philosophical and social significance of our experiment on moral reasoning and the obtained results. This has already been signaled in formula (6) of our article which, according to Immanuel Kant [32], defines the existence of moral norms articulated by the voice of conscience, which declares, in the mode of the categorical imperative, one must act or not act in a certain way in order for a given action to be able to become a universal law.

We are now prepared to carry out a psychological interpretation of the experimental results obtained from the point of view of the assumptions of Kant's philosophy, especially concerning the categorical imperative and the *a priori* structures of the human mind in relation to the moral sphere. The analysis of the indicators of morality constituting the ethical stories according to the scheme of IAO and KCR shows that in each of these schemes the students studied quite easily differentiated the indicators important for the inference of moral judgments from the indicator creating only the situational context for such an assessment. Thus, in the IAO scheme, subjects immediately distinguished the moral inferential indicators (I and A) from the contextual indicator (O), while in the KCR scheme, subjects unerringly recognized the two moral inferential indicators (C and R) as opposed to the contextual indicator (K). This became apparent in the process of integrating just indicators I and A in the conclusion evaluating the entire IAO ethical story, as well as just indicators C and R in the moral evaluation of the KCR schemes.

By perceiving such empirical regularities in the behavior of the subjects under study, one can venture a direction of interpretation that integrates strategies of moral indicators, i.e., logical-syntactic models of moral evaluation of ethical stories in the IAO and KCR schemes are treated as *a priori* forms of the human mind of this moral evaluation. Under this interpretive assumption, the degree of descriptiveness of the logical-syntactic model of the correctness of the moral judgment of each ethical story in the experiment indicates *a priori* its potentiality as a moral regulator of the human mind in situations of moral judgment. On the other hand, the respondents assessing ethical stories in a manner consistent with the model moral evaluations, create a situation that can be defined as a "natural Kantian society"(The authors express their

gratitude to Jeff White for inspiration regarding the interpretation of the obtained results of empirical research in terms of the *natural Kantian society)*.

## Supporting information

**S1 Appendix. Ethical story modeling experiment report form.** https://doi.org/10.6084/m9.figshare.14703495.v1.
(DOCX)

**S2 Appendix. Case study example.** https://doi.org/10.6084/m9.figshare.14703504.v1.
(DOCX)

## Author Contributions

**Conceptualization:** Ján Grác.

**Formal analysis:** Piotr Janusz Mamcarz.

**Investigation:** Dorota Kornas-Biela.

**Methodology:** Adam Biela.

**Visualization:** Dorota Kornas-Biela.

**Writing – original draft:** Ján Grác.

**Writing – review & editing:** Adam Biela, Piotr Janusz Mamcarz.

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
