## [Decision Letter · Decision Letter 0]

4 Jan 2021

PONE-D-20-37673

Can moral reasoning be modeled in an experiment?

PLOS ONE

Dear Dr. Mamcarz,

Thank you for submitting your manuscript to PLOS ONE. After careful consideration, we feel that it has merit but does not fully meet PLOS ONE’s publication criteria as it currently stands. Therefore, we invite you to submit a revised version of the manuscript that addresses the points raised during the review process.

The three reviewers have provided constructive and detailed comments. It is in agreement that the work has merit and could lead to an interesting contribution to PLOS ONE. However, there are several aspects of the paper that need significant improvements. Please carefully consider them in the revision of your manuscript.

We look forward to receiving your revised manuscript.

Kind regards,

The Anh Han, Ph.D.

Academic Editor

PLOS ONE

Journal Requirements:

2. Please consider changing the title so as to meet our title format requirement (https://journals.plos.org/plosone/s/submission-guidelines). In particular, the title should be "Specific, descriptive, concise, and comprehensible to readers outside the field" and in this case it is not informative and specific about your study's scope and methodology.

3. Thank you for including a copy of your questionnaire in English. In accordance with PLOS' guidelines on reporting and reproducibility (https://journals.plos.org/plosone/s/criteria-for-publication#loc-3, https://journals.plos.org/plosone/s/criteria-for-publication#loc-7), please also include a copy of the original version as Supporting Information.

4. PLOS ONE does not copy edit accepted manuscripts. Please proofread for typos and grammar.

5.We note that you have indicated that data from this study are available upon request. PLOS only allows data to be available upon request if there are legal or ethical restrictions on sharing data publicly. For more information on unacceptable data access restrictions, please see http://journals.plos.org/plosone/s/data-availability#loc-unacceptable-data-access-restrictions.

6.Thank you for stating the following financial disclosure:

 "NO"

7.Thank you for stating the following in your Competing Interests section: 

"NO"

9. Please upload a copy of Supporting Information APPENDIX 2 which you refer to in your text on page 31.

Additional Editor Comments (if provided):

The three reviewers have provided constructive and detailed comments. It is in agreement that the work has  merit and could lead to an interesting contribution to PLOS ONE. However, there are several aspects of the paper that need significant improvements. Please carefully consider them in the revision of your manuscript.

Reviewers' comments:

Reviewer's Responses to Questions

**Comments to the Author**

1. Is the manuscript technically sound, and do the data support the conclusions?

Reviewer #1: Partly

Reviewer #2: No

Reviewer #3: Partly

2. Has the statistical analysis been performed appropriately and rigorously? 

Reviewer #1: No

Reviewer #2: No

Reviewer #3: N/A

3. Have the authors made all data underlying the findings in their manuscript fully available?

Reviewer #1: Yes

Reviewer #2: No

Reviewer #3: Yes

4. Is the manuscript presented in an intelligible fashion and written in standard English?

Reviewer #1: No

Reviewer #2: Yes

Reviewer #3: Yes

5. Review Comments to the Author

Reviewer #1: The trouble with the paper is that it is not coherent, as a paper. There are some interesting parts, mostly that second half of the paper. This is written well, and makes sense, at least. Short advice: cut the first 15 pages, and recompose an introduction grounding the IAO and KCR etc. Then, rewrite the stuff from 16 onwards accordingly. The current introduction is a flurry of references, with which few are substantively engaged. Yes, Piaget and Kohlberg are important thinkers, but why in this context are their references so important? This is is not established, only presumed. At the same time, necessary references are missed, and important assertions are oddly not grounded. Where there are references, these are mostly "drive-by" style. There is little substantial and fruitful engagement with any of these works. Though a "drive-by" reference can be useful sometimes, in general, references should ground assertions in common literature, and in specific ways that distinguish the present reseach from others. The basic idea of the first half of the paper should be to focus in on the second half. However, the first half of the paper doesn't do that, and in the end it is not clear that the first half of the paper is necessary, at all. See the attached edited draft PDF for comments. Some comments from the PDF: "One sentence paragraphs are to be avoided." (page 2) "Are these points in time, or periods in the more or less normal biological development of human beings?" (page 3) "Maybe this is a better first sentence for this paragraph>" (page 5) "Can you write this more simply, perhaps as multiple sentences?" (page 5) "Task in?" (page 5) "Certainly? This is not clear. And moreover, you do not explain what these terms mean, and why they may be important for others." (page 9) "Now, this use of “moral” is sensible, as a distinction from what may be cnsidered the “ethical” or social, here your “external”. However, it is not clear that Grac is originally responsible for such a distinction, and still, the three special terms have not been clarified." (page 9) "As your study relies on the work of Grac, the preceding discussion should set this theory out in greater detail. Rather, now the methods section begins by assuming that this is an adequate theory, with which the reader is adequately acquainted. However, the introductory pages are deficient in this regard." (page 10-11). "This doesn’t sound much like Kant’s CI." (page 13) "Does Kant base his CI in personhood?" (page 14) "“precisely stated” - You might cite and perhaps quote relevant examples." (page 14) "Why would these be “weaker”?" (page 15) "Though there is some semblance of a review, there is no systematic exposure of “indicators”." (page 15) "Are these lifted from those resources, or are they products of your assay, or…? " (page 15) "Advice: recompose the first 15 pages as 8 or 10, and better introduce Grac and this approach in the same." (page 16) There are more comments on the PDF, but I will stop here. Ultimately, this would be my advice. Whatever happens, the paper should directly introduce exactly those ideas making the focal experiment and its results transparently significant and informative. Right now, the paper does not do that. Note that to get the current paper to do that qwould require radical re-composition and is in no way an acceptance of the paper for publication. The paper as it has been submitted is clearly not publishable, and I do not see it easily becoming so. Rather, this reviewer feels that there are redeeming aspects to the paper. The authors appear to have taken themselves seriously enough to present the paper, but as a single paper it fails and this reviewer sees no way to make it into a coherent paper within any practical period of time. Some parts of the paper are more easily dismissed than others, for instance, too many controversial claims remain unsubstantiated, including some regarding the moral language specific to certain linguistic groups and what leads from that, i.e. an apparent moral superiority. Why this is necessary in this paper is a mystery to this reviewer. Nothing is done with the references, and the meanings of the terms are never revealed. Still, if it is necessary, then this necessity should be supported in argument. This is not done, and moreover the discussion doesn't add to the paper. Many other sections seem similarly to detract from the paper rather than add. Other big problems are the English grammar throughout the first half, of course. It needs to be proofed by a patient English language editor before resubmission. it is not publishable as presented. There are in the paper some worrisome assumptions (some noted in the PDF, search "assume"). And, there are some problem over-generalizations. As it stands, this is not Q1 research. It might be radically revised and resubmitted, but as a new submission. I would be willing to see such a resubmission. This would - for me - have to better set up and better interpret materials beginning at about page 15 of the present paper. As for the rest, my advice would be to break it into sections, determine what each one is trying to say, and decide how and what it contributes to the message of the paper, overall. As it is, the paper does not show us how its most interesting parts have anything special to do with Piaget, or why the formaulaic Venn representations are useful, ... and more. My advice as a writer and advisor to students would be that, if something does contribute to your message, then you need to show us why. If you can't do that, then you should cut that part. Given the amount of work that I would expect necessary to adequately revise this paper in a timely manner, this reviewer recommends rejection of the present submission with encouragement to resubmit after radical revisions. Although, this reviewer does not expect that the authors will be able to undertake adequate revisions in a reasonable time, ultimately, deadlines depend on the timeline established by the publishers and editors of this issue. With a suitable extension, such radical revisions may be possible.

Reviewer #2: The paper proposes an experimental method on moral reasoning with a control over multi-indicators such as intention, concordance, application, outcome, knowledge, and realization; each with three possible values (+, -, ∅). The method consists of two phases: the creation of model ethical stories and an overall assessment of each of these stories. The moral assessment of these stories, in order to uncover the regularities, are determined by three processes: selection, differentiation, and integration. As the result of applying these three processes, seven regularities are identified, providing a pattern in the form of implication, whose premises are indicators and their corresponding values and the conclusion is the overall moral evaluation of the constructed ethical stories. Such a pattern can thus be applied in moral inference as in the formal logical inference of modus ponens.

The paper is well-written in English (despite some inconsistencies, detailed below) and the whole content can be generally followed. The introduction details existing works in moral reasoning. To some extent, I find that the introduction is too wide in its coverage and it could focus more on existing works on similar experimental methods so as the contribution and the novelty of this paper can easily be distinguished from the existing ones.

In terms of materials and methods, I really appreciate that the methods are written systematically, built from the basic ingredients from introducing class norms to enumerating all configurations for constructing ethical stories. Nevertheless, here are some comments that may improve the paper:

- The idea of mathematically formalizing the method by using sets is a good one. But the way it is formalized is rather non-standard and thus can be confusing if this paper is read by mathematically-related researchers (e.g., computer scientists), e.g., if they want to computationally model your results. One confusion comes from mixing up the logical conjunction notation (∧) and set intersection (∩). Though they are closely related, their use should be distinguished. At first, the paper seems to choose set notation (e.g., intersection for conjunction), but then in formula (6), the logical disjunction notation (∨) is used. Mixing this up renders some formalization superfluous, e.g., (3) - (5) and others. I would suggest all formulas to be rewritten in accordance to a standard formal notation.

- One concern, and this is perhaps because the paper is unclear on the procedure, is whether the same individual subject conducts the first and the second phase. Or, whether these two phases are performed by different individual subjects? If both phases are performed by the same individual subject, my concern is on potential bias in the assessment of the stories (given that the subject is also the one who constructs the stories). This may need clarification in the paper as the integration process relies on the results of the second phase. Perhaps a kind of cross validation in conducting the second phase (evaluation by different subjects, run several times) can be considered as a procedure in order to derive more convincing regularities.

- While the IAO schema is supplemented by matrices that confirm the regularities, unfortunately this is not the case for the KCR schema. I just wonder whether there is no anomaly at all in both schemas and this concern is actually related to my previous point on the procedure. The matrices themselves are not sufficient, but data and relevant statistical analysis from the experiment (particularly from phase 2) are required to justify the seven regularities that come from the experiments.

Minor comments:

- In page 11, the first subclass that is formalized in (2) is not a subclass of CN (formula 1), since CN only contains objects from the intersection of M with E or L. The first subclass requires that it belongs only to M and does not belong to E or L.

- Formula 7, it should be IAO instead of IAR.

The third regularity of the IAO schema is called “the principle of preferring the intention indicator (I) over the *non-zero application indicator (A)*. But Table 9 shows that this principle comes from cases where the application indicator (A) is always zero.

- Formula 19 is inconsistent with the regularity described: the premise should be C≠R instead of I≠R.

Reviewer #3: The paper reports the results of an experiment where 147 educational psychology students were asked to create model ethical stories and then make an overall moral assessment of each of these stories.

The stories fall into two cognitive schemas: IAO and KCR.

IAO schema

I = Intention. This is understood as a motive or intent of the individual’s ethical behavior.

A = Application. This answers the question of whether the intention which is moral is or is not obtained by moral means.

O = Outcome is a possible or an accompanying outcome. The outcome can be understood in the given scheme exclusively as non-causal, i.e. as a consequential circumstance which is related to cause and effect, but which cannot be identified with it.

KCR schema

K = Knowledge. This refers to knowledge about moral norms, which regulates human behavior in a given situation.

C = Concordance. This means the actor’s personal attitude to the particular moral norm regulating human performance in this situation.

R = Realizing. This refers to whether and how the requirements of the moral norm regulating human behavior are fulfilled.

In the article there are some examples of stories fitting these schemas.

An IAO example

I (+) John found a new flat to create better conditions for his siblings to study.

A (-) He did it regardless of the disapproval of his mother, who felt almost mortally hurt by his decision.

O ( Ø ) After having moved to his new flat John found out that the signal for his cell phone was the same quality as in the previous location.

A KCR example

K (+) Igor knew that cruelty to animals is immoral.

C (-) He was not opposed to ill treatment of animals.

R ( Ø ) However, he has never even been in a situation where he watched cruelty inflicted on animals by others.

Each value in the schemas has three possible answers (+ - Ø). Thus each schema has 27 possible combinations (3 x 3 x 3)

In the experiment, the students have been asked to create (or “model”) stories that fit into these two cognitive schemas (IAO and KCR). Then they have been asked to evaluate these stories.

Examples of the actual stories and their evaluations are presented in Appendix 1.

An IAO story

I (+) When Jack got his driving license, he decided that he would never drive a car after drinking alcohol.

A (-) Once, when he was drinking with his friends, he picked up a call from his mother asking him to go pick her up at the station.

O (+) When he came to the garage to get the car, he noticed that his father had already gone to get his mother.

The overall moral evaluation of the ethical story is: [ - ]

A KCR story

K (-) Isidore, a candidate for psychology, did not know about confidentiality in counseling under the psychologist's code of ethics.

C (-) Isidore is against keeping secrets because, according to him, everyone has a right to information.

R (Ø) He has not completed his psychological studies, as it is forbidden for a psychologist to pass on such information to third parties.

The overall moral evaluation of ethical story is: [ - ]

This “two-phase design” is the novelty of the approach of the authors: (1) modelling of ethic-dilemma stories based on logically structured cognitive schemas, and (2) overall ethical evaluation of the modeled morality schemas.

This process produced a database of 536 stories.

The authors claim some patterns emerge from this analysis such as the principle of univocal values of intention and application indicators, the principle of preferring application (A) indicator over intention indicator (I) and the principle of preferring the intention indicator (I) over the non-zero application indicator (A) in the IAO schema.

There are four patterns in the KCR schema. Formalisations of these seven patterns are provided.

Major Points

I like the general idea of doing experiments to analyse moral stories and to investigate exactly what is involved in moral reasoning. I also like the idea of formalising patterns in moral reasoning. So I commend the authors for their work in this area. My view is that the paper should be published with some minor revisions.

I feel rather uneasy about the IAO/KCR schemas. It seems to me these schemas are a somewhat limiting template to impose on moral stories.

Consider this well-known moral problem, Switch. One might write the classic “trolley problem” thus:

1. When Bill joined the railway, he decided he would keep everybody safe and kill or harm no one.

Comment: It seems to me this could be K or I. K would result from reading the workplace health and safety manual. I would be Bill’s intention to do what the safety manual said.

2. Once Bill was standing by the switch, when a runaway trolley raced towards him. Five of his fellow workers were in the main line tunnel. One was in the branch line tunnel. All Bill could do was a) nothing or b) throw the switch. If he did nothing five would die. If he threw the switch one would die.

Comment: It seems to me this could be a story about means and ends and so (at a stretch) a question of Application.

3. Bill threw the switch and killed one to save five.

Comment: This is a causal result (so not O = outcome as I described by the authors). At a stretch it could be Realizing.

Thus, as I analyse it, Switch does not obviously fit into either of the IAO or KCR schemas. To be candid, I think I am “shoehorning” Switch into the IAOKCR categories somewhat. It could be IAR or KAR but this observation alone does not invalidate the experiment. It merely suggests there are more experiments and observations that could (and should) be done. However, I am concerned that having three elements in the schema is a somewhat arbitrary choice. Some language dealing with these objections (if only to say more experiments like this need to be done) would be a good idea.

One the other hand, I found the paper interesting to read. Even though I ended up disagreeing with much of it, this alone is not a reason not to publish it. Morality is massive complex and inherently controversial. More experimental work of this pattern should be done and variations on the themes raised in the paper should be attempted.

Minor Points

If the students have been "carefully instructed" to write stories with the IAO and KCR schemas in mind, how can the experiment be said to be natural? It would seem to be heavily prompted to me. It might be a better idea to have students apply the IAOKCR schemas to well-known moral dilemmas from the ethics literature or to student-authored dilemmas, where the students are not prompted to express a story in a restrictive schema like IAO or KCR. However, this would be a quite different research project. In a revision I would simply avoid the claim the experiment is “natural.”

The difference between executive, moral and legislative norms was a little unclear to me. What an executive norm is should be clarified in a revision.

I was puzzled on the use of the term "moral norm" which sometimes in the paper seemed to refer to a character trait (i.e. a “virtue” or disposition to act) rather than a normative rule or a moral principle. Courage is certainly a “virtue” on Aristotle’s list. I would use virtue instead or character trait rather than moral norm in these cases.

References to GE Moore were incomplete (not all references in the text appeared in the References section). Also statements made about Moore’s ethical position were somewhat inaccurate. It is not quite right to describe GE Moore’s philosophy as emotivism. Emotivism is more closely associated with figures such as AJ Ayer and CL Stevenson and is the view that moral sentences such as “murder is wrong” express speaker’s feelings rather than objective truth. The claim is that such sentences do not refer to anything “objective” and thus are not truth apt. Emotivism is thus a non-cognitivist moral theory.

It is true that GE Moore’s attack on ethical naturalism did give rise to non-cognitivist moral theories such as emotivism, prescriptivism and expressivism but Moore is a moral realist and a cognitivist who holds the “good” is a non-natural property that can be directly perceived by people. He holds the “naturalistic fallacy” results from trying to equate “good” with a natural property such as “pleasure” but this is not the extreme “boo/hooray” view of morality found in AJ Ayer’s emotivism.

The characterisation of the Categorial Imperative provided by the authors was a little too distant from Kant’s three formulations to be accurate. I would drop the reference to Kant and find another term for Categorial Imperative in the formalisation.

The article mentioned three Slovak words mravnost’, moralnost’ and etika. Some explanation of the difference in Slovak between mravnost’ and moralnost’ should be offered for the benefit of non-Slovak speakers. Alternatively, the point could be cut as it does not seem essential to the argument made in the paper.

6. PLOS authors have the option to publish the peer review history of their article (what does this mean?). If published, this will include your full peer review and any attached files.

Reviewer #1: **Yes: **Jeffrey White

Reviewer #2: No

Reviewer #3: No

---

## [Author Response · Author response to Decision Letter 0]

19 Feb 2021

ANSWERS FOR THE REVIEWERS

For Reviewer #1

Your comments were, indeed, a well-chosen remedy for the first part of the article. I treated them as good indicators for improving the internal consistency of this part of the study in terms of its substantive coherence. Hence, at your suggestion, Part One has generally been reorganized as follows: 

1. The main goal of the paper is stated at the very beginning, i.e. the presentation of Jan Grác’s (2008) experimental approach to the study of moral reasoning with the possibility of controlling some indicators of morality as independent variables. This methodological approach is presented first in the context of the classic approaches to moral reasoning, and then in the latest contemporary research, with particular emphasis on the possibility of experimental exploration of this research field.

2. After pointing to some contemporary authors known in the literature on the subject, I turned my attention to Piaget and Kohlberg as the classic authors of research on moral reasoning. At the same time, I indicated the relationship between their concepts and methodologies and Grác’s proposed methodology.

3. At the same time, a significant number of paragraphs have been completely removed from the text of the article as being inconsistent with the main goal of our analysis.

4. I consider the meta-analysis to be an essential part of the introduction. However, I have limited myself (in accordance with your recommendation) to relations with those items which directly concern experimental research on moral reasoning.

5. After these preliminary analyses, J. Grác's theoretical and methodological statements on moral reasoning are presented in more detail (pp.10-14 of the reedited manuscript).

6. Regarding your comments on Kant's CI, I have decided to refer to his fundamental work in this area for his own definition (E. Kant (1966). Groundwork of the Metaphysics of Morals, in I. Kant, Practical Philosophy, translated by Mary Gregor, Cambridge; Cambridge University Press, Chapter 4, page 421).

7. Regarding the “no systematic exposure of indicators,” the corresponding paragraph in the text has been reformulated.

8. The use of the Venn diagram in the analysis of the range relations between three different names of standards (moral, legal, executive) is justified by the necessity to strictly distinguish them in research practice. In this spirit, an appropriate paragraph explaining this issue was added to the text. 

For Reviewer #2

1. As the first remarks of Reviewer # 2 are in harmony with the remarks of Reviewer # 1, we respond that the introductory part has been reworked towards making it more coherent and shorter.

2. Thanks for your attention regarding mixing the use of symbols in formalization formulas. In the end, I decided to standardize the symbolism in accordance with the notation of set theory, as these formalizations basically relate to the range relations between set names designates (e.g. moral, legal, executive norms in a Venn diagram, or in moral evaluation, which is also based on set operations).

3. You are right about the ambiguity of the experimental procedure presented in the first version of our manuscript and of the tested persons. Your objections on this matter are of vital importance. Hence, our work required clarification of such issues as: 1) the studied groups of students; 2) specification of which part of the experiment they participated in; and 3) presentation of the results of research on the strategies used by the respondents to evaluate ethical stories in the second part of the experiment. For this reason, the authors of this study decided to select two other groups of young people applying to study psychology for the second part of the experiment, i.e. for overall evaluation of the ethical stories (modeled in the first part of the experiment) as subjects. They had just completed secondary school and were taking their entrance exams for psychology at the University of Trnava. These subjects are called Subject Group 2.Their number is - 450 (N=450). The relevant fragments entered in the manuscript are marked in red.

4. Your comments related to the Formula (1). I decided to precise the notion of this formula as follows: 

 CN = {E, M, L, (E ⋂ M), (L ⋂ M), (E ⋂ L) (E ⋂ M ⋂ L)} 

I am operating with the symbol Ṁ ( and also with M) which denotes the subclass of moral norms that do not belong conjunctively either to executive norms or to legislative ones. This symbol refers to the moral norms which I have called pure or context-free moral norms. I have marked these items in red. The symbol M in our analysis denotes the universal class of moral norms.

5. Thank you very much for the very insightful remarks on formal defects in the records of some symbols (as in Formula 7 and formulations concerning the third regularity of general moral evaluation according to the third rule of the IAO scheme). Appropriate adjustments have been made.

For Reviewer # 3

1. Your first critical remark concerns whether the IAO / KCR schemes used in the experiment are a kind of limiting template imposed on ethical stories. You also gave interesting and inspiring comparisons of our experiments with other paradigms of research on moral reasoning. An example may be the classic "trolley problem" with references to indicators of morality in moral schemes of the IAO / KCR. Thank you for this suggestion, as the classic "trolley problem" could be positioned within the moral framework of our experiment in future research.

It is true that in every experiment the research scheme, often called a paradigm, and in our experiment a cognitive schema (IAO and KCR), limits the freedom of the experimenter and the subjects. This is the downside to all experiments by definition. However, this limitation ensures in our experiment both the possibility for the experimenter to control the independent variables (IAO morality indicators in one scheme and KCR indicators in the other scheme) as well as the precision of this control by significantly reducing the contaminating influence of uncontrolled factors on the dependent variable – which is the general moral evaluation of the ethical story. As you noticed in your reporting part of the review, the design of the experiment provides for precise control within one cognitive schema, combinatorically defined 27 values of three indicators of morality. Due to the bounded rationality of the human mind, more indicators of morality cannot be included in one framework. The selection of only two morality indicators for experimental control gives the possibility of precise control of 9 values of these variables (+, -, O), but the two-indicator moral schemas seemed to have too little context. Therefore, it was decided to choose Aristotle's "golden mean" in our experiment. 

2. We have simply avoided the claim that our experiment is “natural.” Your arguments convince us.

3. What characterizes executive norms? They are unambiguous regulators of the behavior manifested in specific situations, e.g. in contact with nature, with other people, with oneself. They are oriented towards the hic et nunc point of view, which is the optimum performance of certain activities. For example, in a sport, a sports record can be quickly replaced by another sports record (ski jumping hill record marked in green to be visible to the jumpers). Another clear feature of this type of normative regulation is that although they improve human activities and accompany various mental experiences (e.g. attitudes, motives), the content of these experiences themselves is not subject to assessment by these norms. Thus, it can be said that neither the views nor the feelings accompanying the performance of these activities are subject to regulation within the framework of moral norms.

4. Your critical remark concerns the use of the term "moral norm" in two different senses: (a) as a virtue (e.g., courage) in the sense of Aristotle's list of virtues; and (b) as a normative rule or moral principle. We would like to clarify that in the first sense we are referring to moral content by basically discussing some items from the subject literature in the introductory part of the article. In its revised version, many of these items have been abandoned and therefore are not included in the text. However, we cannot take advantage of your suggestion to replace the meaning (b) of the term with the meaning (a), because it is fundamental to Grác’s concept and necessary in the conceptual-methodological system of our entire experiment.

5. We agree with the comments on G.E. Moor, and therefore we have excluded these text fragments from the revised manuscript as unrelated to the merit of this article.

6. In our revised version of the article you can find the appropriate formulation of Categorical Imperative based on Kant’s (E. Kant (1966) Groundwork of the Metaphysics of Morals, in I. Kant, Practical Philosophy, translated by Mary Gregor, Cambridge; Cambridge University Press, Chapter 4, page 421).

7. As far as the concepts “mravnost” and “moralnost” are concerned, a special short paragraph has been prepared which synthetically explores the novelty of these concepts.

---

## [Decision Letter · Decision Letter 1]

22 Mar 2021

PONE-D-20-37673R1

Can moral reasoning be modeled in an experiment?

PLOS ONE

Dear Dr. Mamcarz,

Thank you for submitting your manuscript to PLOS ONE. After careful consideration, we feel that it has merit but does not fully meet PLOS ONE’s publication criteria as it currently stands. Therefore, we invite you to submit a revised version of the manuscript that addresses the points raised during the review process.

ACADEMIC EDITOR: The two reviewers have provided further comments on the revised version. It is agreement that improvements have been made and the work has good merit. However,  there are still several issues of the paper that require addressing. Please carefully consider them in the revision of your manuscript.

We look forward to receiving your revised manuscript.

Kind regards,

The Anh Han, Ph.D.

Academic Editor

PLOS ONE

Additional Editor Comments (if provided):

The two reviewers have provided further comments on the revised version. It is agreement that improvements have been made and the work has good merit. However, there are still several issues of the paper that require addressing. Please carefully consider them in the revision of your manuscript.

Reviewers' comments:

Reviewer's Responses to Questions

**Comments to the Author**

1. If the authors have adequately addressed your comments raised in a previous round of review and you feel that this manuscript is now acceptable for publication, you may indicate that here to bypass the “Comments to the Author” section, enter your conflict of interest statement in the “Confidential to Editor” section, and submit your "Accept" recommendation.

Reviewer #1: (No Response)

Reviewer #3: All comments have been addressed

2. Is the manuscript technically sound, and do the data support the conclusions?

Reviewer #1: Partly

Reviewer #3: Yes

3. Has the statistical analysis been performed appropriately and rigorously? 

Reviewer #1: N/A

Reviewer #3: Yes

4. Have the authors made all data underlying the findings in their manuscript fully available?

Reviewer #1: Yes

Reviewer #3: Yes

5. Is the manuscript presented in an intelligible fashion and written in standard English?

Reviewer #1: Yes

Reviewer #3: Yes

6. Review Comments to the Author

Reviewer #1: There are some big problems with the paper, as it is now, and these forbid its recommendation for publication in its current form.

One problem is the specious use of resources, for instance the use of Sternberg, the use of Cushman, the use of Biela, the use of Sudic and Ciric, and for example the use of von Grundherr which is a paper about school bullying from 2017 used in reference to "Analyzing the logical foundations of Kohlberg's research paradigm" on page 5 of the review PDF. In the end, the scholarship is not at a standard that this reviewer can recommend for publication.

Another problem, related with the first, is that the paper does not tie into established and ongoing research well. It goes out of its way to suggest the "novelty" of Grac's research, with most reference to a rather obscure (for most readers) Slovakian publication. At the same time, there is little effort to relate this work with other work in complementary areas, besides some rather erudite 'grounding' in Kohlberg, for instance, and some pointing to some possibly related work without these relationships ever becoming significant for the paper or its results. As such, the paper does not make its importance clear to the reader, perhaps outside of the general approach, because these relationships are not provided.

A third problem has to do with the paper as a paper. It does not hang together well, for one thing, with reference to Kant then none, and like this many ideas come and pass by but are not well integrated. The paper has a "conclusion" and a "final remarks" section... In the conclusion, there is substantial discussion, introducing new ideas and new resources right up to the final paragraphs. This is typically not the role of a conclusion in a technical paper. This is a structural problem, but these problems with the integrity of the paper as a paper go farther than such road-map issues. For instance, how is Dewey and conscience important, exactly? This comes, and is forgotten in the paper, but why bother running through these concepts if only to reduce them to the internalization of norms? This may have been a part of Grac's original thinking, but why should we think about conscience here? Because conscience is educated?

There are other reasons for rejecting this paper for publication, too many to list. The abstract, for example, begins with a 'paragraph' that is one sentence long. It ends by suggesting that experimental results herein described can be useful in education. How is this the case? The issue is not considered again, even in conclusion, so there is a feeling of disjunct, here, and this contributes to the sense that this paper is not one integrated whole developed thought, and rather still requires updating to bring it into context of relevance for contemporary researchers. Then, there is the use of terms of art and technical terms, like "cognitive", or the "multidimensional field of moral reasoning", and phrases like "moral reasoning is a multi-faceted phenomenon where its mutually linked factors constitute the moral existence of the individual human person". Sure, such expressions are understandable, but not exactly clear. And after all that, this statement iabout moral reasoning is not an obvious summary of "above research findings" which include considerations about self-driving cars, for example.

Finally, I have added some comments to the PDF of the draft (attached). Authors may attend to, for example, comments such as on page 4 of the text, concerning a short single-sentence paragraph pointing to "new authors" that only leads to immediate discussion of work from 1973. it is from here that the paper jumps to grounding Kohlberg's "logical foundations" in a paper on school bullying. Why is this paper relvant here, exactly? This should be made clear, or the reference removed.

In the end, it is too much to expect of a reviewer that s/he attend to every infelicity in a paper like this one. The logical notation may be readable, and easily, mostly used to restate what is given in Venn diagrams. These sections are especially plodding, and this speaks to a general consideration. In general, I cannot recommend a paper for publication that I could not also recommend to a student or colleague interested in such issues. Though the general approach may bear some merit, I cannot imagine an situation in which I would recommend that anyone read this paper. Is there something worthwhile in the paper? I think yes. But, it is not presented in a form that I can recommend for publication.

Reviewer #3: I am satisfied all the points I raised have been addressed.

I thank the authors for enlightening me on the difference between mravnost and moralnost. The former seems to point more precisely to the idea of "virtue is its own reward" whereas the latter points to "going through the (moral) motions" or obeying moral rules without appreciating them or even believing in them. I would hesitate to accept the difference cannot be articulated in other languages such as English - even though you need a phrase not a single word to do it. Personally I would be disinclined to draw the conclusion "This certainly means that psychologically, the users of these two languages operate with more developed cognitive and behavioral structures for the moral domain than the users of other European languages" which could be interpreted as a claim of Czech and Slovak moral supremacy!

People with a thinner skin than this reviewer might be offended. Many such people exist in these febrile times! It seems to me the entire discussion is tangential to the main line of argument and could be cut. However, I leave this decision to the authors.

7. PLOS authors have the option to publish the peer review history of their article (what does this mean?). If published, this will include your full peer review and any attached files.

Reviewer #1: No

Reviewer #3: No

---

## [Author Response · Author response to Decision Letter 1]

27 Apr 2021

For Reviewer #1

1. Thanks for your immense work on our manuscript, which is evident both in formal review comments and particularly in your red color comments on the PDF manuscript version. What did we do? We decided to make our manuscript more coherent and took your points of criticism concerning the presence of J. Dewey and I. Kant in our paper more seriously. 

2. You can see from the actual version of the manuscript that J. Dewey and I. Kant are now discussed in our paper after your inspirative remarks.

For Reviewer #3

1. We are also grateful to Reviewer 3 for a creative discussion of the concepts of mravnost and morality and for making his own suggestion as to the distinction between the two. We have taken the liberty of securing these comments in an appropriate footnote in our article.

2. After considering your points about the evaluation sentence regarding Slovak and Czech language users, this sentence has been removed from the text of the article.

---

## [Decision Letter · Decision Letter 2]

13 May 2021

PONE-D-20-37673R2

Can moral reasoning be modeled in an experiment?

PLOS ONE

Dear Dr. Mamcarz,

Thank you for submitting your manuscript to PLOS ONE. After careful consideration, we feel that it has merit but does not fully meet PLOS ONE’s publication criteria as it currently stands. Therefore, we invite you to submit a revised version of the manuscript that addresses the points raised during the review process.

ACADEMIC EDITOR: The paper has been significantly improved with the revisions. There remain some minor issues that are required to ensure that all the conclusions are supported by the data. Please carefully consider the reviewer’s suggestions in the revision of your manuscript.

We look forward to receiving your revised manuscript.

Kind regards,

The Anh Han, Ph.D.

Academic Editor

PLOS ONE

Journal Requirements:

Additional Editor Comments (if provided):

The paper has been significantly improved with the revisions. There remain some minor issues that are required to ensure that all the conclusions are supported by the data. Please carefully consider the reviewer’s suggestions in the revision of your manuscript.

Reviewers' comments:

Reviewer's Responses to Questions

**Comments to the Author**

1. If the authors have adequately addressed your comments raised in a previous round of review and you feel that this manuscript is now acceptable for publication, you may indicate that here to bypass the “Comments to the Author” section, enter your conflict of interest statement in the “Confidential to Editor” section, and submit your "Accept" recommendation.

Reviewer #1: (No Response)

2. Is the manuscript technically sound, and do the data support the conclusions?

Reviewer #1: (No Response)

3. Has the statistical analysis been performed appropriately and rigorously? 

Reviewer #1: (No Response)

4. Have the authors made all data underlying the findings in their manuscript fully available?

Reviewer #1: (No Response)

5. Is the manuscript presented in an intelligible fashion and written in standard English?

Reviewer #1: (No Response)

6. Review Comments to the Author

Reviewer #1: When consulting the guidelines for publication in this journal, one finds difficulty only with number 4: Conclusions are presented in an appropriate fashion and are supported by the data. In general, claims are supported. New additions which make clear the relationship between experimental findings and Kantian moral reasoning go a long way to bringing the paper together, in the end. One claim, however, stands out as unsupported by data/argument, and this is the following: "Thus, we may say that this experiment took place in closer proximity to real-life thinking processes than any other research method previously used in experimental psychology." For one thing, this "Thus" is the first word of a new paragraph, when generally "thus" and "therefore" for example are used at the ends of paragraphs, to indicate the conclusion of an argument rather than the beginning of a new one. But most importantly, this "any" is hyperbole. The assertion distinguishing the present research from "any other research method previously used in experimental psychology" is not established by sufficient review of extant psychological literature, at least not in this paper. The paper does not establish this assertion, clearly, in fact. Thus, in consideration of guideline 4, this claim must either be substantiated, deleted as hyperbole, OR modified to accord with what this paper is able to establish (perhaps that experimenters were sensitive to ecological validity in the design of the experiment, and that future experiments may benefit through adoption of a similar method, ... something like this may be warranted).

Recent revisions improve the paper, and the current presentation is professional enough. Though some claims remain ... excessive, personally ampliative, over-reaching,... these are matters of style and clarity that may impede reader interest but maybe not publication. In the end, this depends on competitiveness of submissions and expectations of editors. If the changes above specified are exacted, given publications guidelines for this journal, this paper can be made publishable with minor revisions.

7. PLOS authors have the option to publish the peer review history of their article (what does this mean?). If published, this will include your full peer review and any attached files.

Reviewer #1: **Yes: **Jeff White

---

## [Author Response · Author response to Decision Letter 2]

15 May 2021

ANSWER TO REVIEWER#1

1. The authors express their gratitude to Reviewer # 1 for inspiration regarding the interpretation of the obtained results of empirical research in terms of the natural Kantian society. This has been expressed in footnote # 5.

2. In the last part of our paper, we have made the appropriate changes in the manuscript in accordance with your suggestions.

3. However, we cannot agree with your argumentation where you are claiming that our assertion concerning “any” is of a hyperbolic kind:” The assertion distinguishing the present research from "any other research method previously used in experimental psychology" is not established by sufficient review of extant psychological literature, at least not in this paper.” 

4. We think that our statement can be seen as quite well documented as we considered in our paper, first of all, a monumental meta-analysis by Ellemers et all, 2019 who analyzed positions on the morality of 1940-2020 from where such conclusion can de drown directly. Moreover, our own analysis of articles from the last three years ( e.g. experiments made by Sudić & Ćirić, 2019; by Awad, 2019; and by Rhim, Lee & Lee, 2020) are also supporting the statement on the novelty of the presented in our paper research method.

---

## [Decision Letter · Decision Letter 3]

21 May 2021

Can moral reasoning be modeled in an experiment?

PONE-D-20-37673R3

Dear Dr. Mamcarz,

We’re pleased to inform you that your manuscript has been judged scientifically suitable for publication and will be formally accepted for publication once it meets all outstanding technical requirements.

Kind regards,

The Anh Han, Ph.D.

Academic Editor

PLOS ONE

Additional Editor Comments (optional):

Reviewers' comments:

Reviewer's Responses to Questions

**Comments to the Author**

1. If the authors have adequately addressed your comments raised in a previous round of review and you feel that this manuscript is now acceptable for publication, you may indicate that here to bypass the “Comments to the Author” section, enter your conflict of interest statement in the “Confidential to Editor” section, and submit your "Accept" recommendation.

Reviewer #1: All comments have been addressed

2. Is the manuscript technically sound, and do the data support the conclusions?

Reviewer #1: Yes

3. Has the statistical analysis been performed appropriately and rigorously? 

Reviewer #1: N/A

4. Have the authors made all data underlying the findings in their manuscript fully available?

Reviewer #1: (No Response)

5. Is the manuscript presented in an intelligible fashion and written in standard English?

Reviewer #1: (No Response)

6. Review Comments to the Author

Reviewer #1: Sufficient attention to publication guidelines has been demonstrated. There are no concerns remaining here.

7. PLOS authors have the option to publish the peer review history of their article (what does this mean?). If published, this will include your full peer review and any attached files.

Reviewer #1: **Yes: **Jeffrey White

---

## [Editor Report · Acceptance letter]

2 Jun 2021

PONE-D-20-37673R3 

Can moral reasoning be modeled in an experiment? 

Dear Dr. Mamcarz:

I'm pleased to inform you that your manuscript has been deemed suitable for publication in PLOS ONE. Congratulations! Your manuscript is now with our production department. 

Kind regards, 

on behalf of

Dr. The Anh Han 

Academic Editor

PLOS ONE